# Faster Directional Convergence of Linear Neural Networks under Spherically Symmetric Data

**Dachao Lin**[1]    **Ruoyu Sun**[2]    **Zhihua Zhang**[3]
[1]Academy for Advanced Interdisciplinary Studies, Peking University
[2]Department of Industrial and Enterprise Engineering, Coordinate Science Lab (affiliated)
University of Illinois Urbana-Champaign
[3]School of Mathematical Sciences, Peking University
lindachao@pku.edu.cn   ruoyus@illinois.edu   zhzhang@math.pku.edu.cn

## Abstract

In this paper, we study gradient methods for training deep linear neural networks with binary cross-entropy loss. In particular, we show global directional convergence guarantees from a polynomial rate to a linear rate for (deep) linear networks with spherically symmetric data distribution, which can be viewed as a specific zero-margin dataset. Our results do not require the assumptions in other works such as small initial loss, presumed convergence of weight direction, or overparameterization. We also characterize our findings in experiments.

## 1 Introduction

**Local v.s. non-local analysis.** Deep neural networks have been successfully trained with simple gradient-based methods, despite the inherent non-convexity of the objective function. Recently, a number of works proved the convergence of gradient methods to global minima for ultra-wide neural networks [1, 4, 12, 13, 20, 31]. These works essentially performed "local analysis" because in their proofs the parameters stay close to initialization during training. In the large width setting, there exists a global minimum near a random initial point, thus staying in the local region can still result in convergence to global minima. Despite the technical convenience of handling a local region, in practical training, the parameters often travel far from initialization (see, e.g., [15]). To understand the practical optimization trajectory, it is important to develop a *non-local* convergence analysis.

**Deep linear nets with quadratic loss: "local analysis".** To understand the behavior of gradient descent (GD) for general neural nets, we need to understand GD for deep linear nets. A number of works analyzed linear networks with the square loss. Bartlett et al. [7] provided that gradient descent converges to the target matrix at a linear rate from identity initialization, while assuming the target matrix is either close to identity or positive definite. Arora et al. [2] proved linear convergence of deep linear networks if the initialization has a positive "deficiency margin" and is nearly balanced. Later, a few works followed a similar idea to neural tangent kernel (NTK) [20] to establish convergence analysis. Du and Hu [11] showed GD with Gaussian random initialization converges to a global minimum at a linear rate if the width of every hidden layer is large enough. Hu et al. [19] improved the width requirement to be independent of depth, by utilizing orthogonal weight initialization, but they assume each layer to have the same width. All the above works are not non-local analysis.

**Deep linear nets with quadratic loss: non-local analysis.** Eftekhari [14] provided non-local convergence analysis for deep linear nets with quadratic loss. When one layer has only one neuron (including scalar output case) and assuming that the input data are whitened, they proved that gradient flow converges to global minimizers starting from balanced initialization.

In this work, we are interested in the classification task with binary cross-entropy loss. To our knowledge, there was no non-local convergence analysis of gradient flow for deep linear nets in such a scheme.

**Deep linear nets with logit loss: final-phase analysis.** Several recent works [17, 38, 23, 21, 37] studied the exponential-type loss for deep linear networks. Specifically, Gunasekar et al. [17], Nacson et al. [38] proved the convergence to max-margin solution, but under the assumption that the weight direction and the loss have converged to global optima, which is a "final-point analysis". Lyu and Li [37], Ji and Telgarsky [21, 23] also proved the convergence to max-margin solution under the assumption that the initial point has already obtained zero classification error (weaker assumption than [17, 38]), i.e., they analyzed the "final phase" of training. In this work, we would like to understand the entire training dynamics, not just the "late training" period.

## 1.1 Our Contributions

In this paper, we analyze gradient flow for deep linear networks with logit loss (i.e. binary cross-entropy loss). The main contributions of this paper are summarized as follows:

- **Convergence result.** We prove the global convergence of gradient flow for minimizing a population logit loss with deep linear nets, under the assumption that the input data is spherically symmetric (Theorem 3). This assumption covers the standard Gaussian distribution and uniform distribution on the sphere. To our knowledge, this is the first global analysis of gradient flow on deep linear nets with logit loss, though under certain strong assumptions. We emphasize that our analysis is beyond the "lazy training" scheme.

- **Convergence rate.** We also establish explicit convergence rate of gradient flow (Theorem 3). Denote $\theta(t)$ as the angle between $\mathbf{w}_e(t)$ and $\mathbf{v}$, where $\mathbf{w}_e(t)$ is the collection of parameters at time $t$ following gradient flow, and $\mathbf{v}$ is the directional global minimizer. In the first phase, we have polynomial convergence rate of $\cos(\theta(t))$, that is, $\cos(\theta(t)) \geq 1 - O\big((N_1 t)^{-\frac{2}{N_1 \pi}}\big)$, where $N_1 + 2$ is the depth of the deep linear network. In the second phase, we have linear convergence rate of $\cos(\theta(t))$, that is, $\cos(\theta(t)) \geq 1 - e^{O(-t)}$. And the second phase begins when the induced weight norm changes from descending to ascending mentioned below.

- **Weight norm change pattern.** We prove that the induced weight norm goes through descending and ascending periods. If the initial induced weight norm starts with descending behavior, then after finite time, it will change to ascending and continues increasing to infinity. If the initial induced weight norm begins with ascending behavior, then it would increase to infinity directly.

We also verify our results in numerical experiments including the descending and ascending behavior of weight norm and the convergence rates in our setting.

## 1.2 Additional Related work

**Directional convergence for exponentially-tailed loss.** Previously, we mentioned a few works on deep linear nets for classification. Next, we discuss convergence results on classification for more general settings (not just deep linear nets). For binary classification problems, oftentimes it is the direction of the predictor that matters, while the norm is less important. For a finite number of linearly separable data, there can be infinitely many directions that lead to zero loss, and classical optimization results do not show which one GD converges to. In many recent works [22, 24, 25, 38, 39, 46, 16], it is shown that for binary classification problems with exponential-type loss (which can be viewed as the 1-layer 1-linear-neuron net case) and linearly separable data, GD converges to the $\ell_2$ maximum margin direction. These works assume there is a positive margin. Ji and Telgarsky [22], Ji et al. [25] also consider a non-separable case, but still requires a positive margin in the "separable part". Moreover, some works [21, 23, 17, 38] study the convergence of GD for training deep linear nets. Lyu and Li [37] further studied deep homogeneous networks (covering deep linear nets and deep ReLU nets). Yun et al. [51] provided a unified framework that connects several existing results under a general tensor formulation of linear networks. A few other works study wide non-linear neural nets, e.g., Chizat and Bach [9], Bach and Chizat [5] improved previous results in the infinitely wide two-layer case, identifying the learned classifier as the solution of a convex max-margin problem.

There are two differences between those works and ours. First, none of these works have performed non-local convergence analysis for deep linear nets as the current work does. Second, they consider finite data with a positive margin, and we consider infinitely many data with a zero margin. We underline that previous convergence rates are inversely proportional to such a margin, which is vacuous in our zero-margin setting.

**More discussions on non-local analysis of deep linear nets with quadratic loss.** We mentioned Eftekhari [14] earlier; here we add more discussions. Besides the main difference in the loss functions, there are two other differences between Eftekhari [14] and our work. First, they assume whitened data, while we assume spherically symmetric data. Second, their convergence rate result requires an extra assumption that the inverse spectral gap of the target matrix is small ([14, Theorem 4.4]), while our convergence rate result does not require extra assumption. Besides the work [14], another earlier work [6] performed "non-local analysis", but as pointed out by [14], their work only ensures convergence to the minimizer in a constrained subset, not necessarily the global minimizer (see Appendix F in [14] for detail).

**Global landscape analysis.** In our work, the term "non-local analysis" should actually be called "non-local *convergence* analysis". There is another line of works on "non-local *landscape* analysis" (see, e.g., [26, 36, 27, 42, 52, 41, 30, 10, 34, 33, 35, 49, 44] and the surveys [48, 47]) which analyze the properties of stationary points, local minima, strict local minima, etc. These works do not directly analyze the convergence of GD, thus are different from our work.

## 2 Preliminaries

We consider a binary classification problem where the data are generated as follows. Let $\mathbf{v} \in \mathbb{R}^d$ be a fixed vector satisfying $\|\mathbf{v}\| = 1$. The input data point $\mathbf{x} \in \mathbb{R}^d$ is sampled from a distribution $\mathcal{D}$ and the label for each $\mathbf{x}$ is $y(\mathbf{x}) = \mathrm{sgn}(\mathbf{v}^\top \mathbf{x}) \in \{-1, +1\}$ [1].

Suppose we want to learn the parameter $\mathbf{w}$ for a predictor $\phi(\cdot, \mathbf{w}) \colon \mathbb{R}^d \to \mathbb{R}$ (the specific form of $\phi$ will be discussed later). We use the population logit loss $\ell(z) := \ln(1 + e^{-z})$ (binary cross-entropy) with $z(\mathbf{x}, \mathbf{w}) = y(\mathbf{x}) \cdot \phi(\mathbf{x}, \mathbf{w})$. The parameter is learned by solving the optimization problem

$$\min_{\mathbf{w}} L(\mathbf{w}) := \mathbb{E}_{\mathbf{x} \sim \mathcal{D}} \ln\left(1 + e^{-y(\mathbf{x})\phi(\mathbf{x}, \mathbf{w})}\right). \tag{1}$$

We focus on the following standard gradient methods.

1) **Gradient flow**: We initialize $\mathbf{w}(0)$, and for every $t > 0$ let $\mathbf{w}(t)$ be the solution of the differential equation: $\dot{\mathbf{w}}(t) = -\nabla L(\mathbf{w}(t))$.

2) **Gradient descent**: We initialize $\mathbf{w}(0)$ and set a sequence of positive learning rates $\{\eta_k\}_{k=1}^\infty$. At each iteration $n > 0$, we do a single step in the negative direction of the gradient, that is, $\mathbf{w}(k+1) = \mathbf{w}(k) - \eta_k \nabla L(\mathbf{w}(k))$.

**Notation.** We denote vectors by lowercase bold letters (e.g., $\mathbf{u}, \mathbf{x}, \mathbf{v}$). Moreover, $\|\cdot\|$ denotes the standard $\ell_2$-norm for a vector, or the spectral norm for a given matrix. Let $\Sigma = \mathbb{E}_{\mathbf{x} \sim \mathcal{D}} \mathbf{x}\mathbf{x}^\top$ be the population covariance matrix, and $\lambda_{\max}(A)$ be the maximum eigenvalue of a real symmetric matrix $A$. We by $\mathcal{S}^{d-1}$ denote the surface of an $d$-dimensional unit sphere, and by $\boldsymbol{I}_k$ the $k$-dimensional identity matrix.

We let $\overline{\mathbf{w}} := \mathbf{w}/\|\mathbf{w}\|$ whenever $\|\mathbf{w}\| \neq 0$. Given vectors $\mathbf{w} = (w_1, \ldots, w_d)^\top \in \mathbb{R}^d$ and $\mathbf{v} = (v_1, \ldots, v_d)^\top \in \mathbb{R}^d$, we let $\theta(\mathbf{w}, \mathbf{v}) := \arccos\left[\mathbf{w}^\top \mathbf{v}/(\|\mathbf{w}\| \cdot \|\mathbf{v}\|)\right] \in [0, \pi]$ denote the angle between $\mathbf{w}$ and $\mathbf{v}$, $\theta(t) := \theta(\mathbf{w}(t), \mathbf{v})$ and $\theta(n) := \theta(\mathbf{w}(n), \mathbf{v})$. We let $\mathcal{D}_{\mathbf{w}, \mathbf{v}}$ be the marginal distribution of $\mathbf{x} = (x_1, \ldots, x_d)^\top$ on the subspace spanned by two linearly independent vectors $\mathbf{w}, \mathbf{v}$ (as a distribution over $\mathbb{R}^2$), and $\mathcal{D}_{\mathbf{v}}$ similarly. Let $\mathcal{D}_2 := \mathcal{D}_{\mathbf{e}_1, \mathbf{e}_2}$ where $\boldsymbol{e}_1, \ldots, \boldsymbol{e}_d$ are the $d$-dimensional coordinate directions, and $c_0 := \mathbb{E}_{\mathbf{x} \sim \mathcal{D}_2} \|\mathbf{x}\|$.

Additionally, we use the standard $\mathcal{O}(\cdot), \Omega(\cdot)$ and $\Theta(\cdot)$ to hide universal constant factors. We say the algorithm achieves *superpolynomial* convergence if it takes $\mathcal{O}\left(\ln^\alpha(1/\epsilon)\right)$ iterations (for a constant $\alpha > 0$) to achieve $\epsilon$-error.

---

[1] Here, $\mathrm{sgn}(z) = 1$ if $z \geq 0$, otherwise $-1$.

## 2.1 Data distribution assumption

**Assumption 1** $\mathcal{D}$ *is a spherically symmetric distribution, i.e., if* $\mathbf{x} \sim \mathcal{D}$*, then for any orthogonal matrix $A$, we have $A\mathbf{x} \sim \mathcal{D}$.*

Unlike the common finite dataset, our main assumption on the data is the above. Such construction gives a zero-margin separable dataset in our scheme. Moreover, our assumption includes the standard Gaussian distribution (adopted in [8, 53, 45, 54, 32]), and uniform distribution on a sphere $\mathcal{S}^{d-1}$. Under this assumption, $\mathcal{D}_{\mathbf{w},\mathbf{v}}$ shares the same distribution for different pairs of $\mathbf{w}, \mathbf{v} \in \mathbb{R}^d$ as long as $\mathbf{w}$ and $\mathbf{v}$ are not parallel (i.e., $\mathbf{w} \nparallel \mathbf{v}$).

# 3 Warm-up: Logistic Regression Case

In this section, we analyze the logistic regression setting, i.e., $\phi(\mathbf{x}, \mathbf{w}) = \mathbf{w}^\top \mathbf{x}$. All proofs of this section are given in Appendix A. For the linear classifier, we have the gradient expression as follows

$$\nabla L(\mathbf{w}) = -\mathbb{E}_{\mathbf{x}\sim\mathcal{D}} \frac{y(\mathbf{x})\mathbf{x}}{1 + e^{y(\mathbf{x})\mathbf{w}^\top\mathbf{x}}}. \tag{2}$$

## 3.1 Convergence of gradient flow for logistic regression

Our main result about logistic regression is the following. The proof is given in Appendix A.6.

**Theorem 1** *Consider gradient flow for solving the problem* (1)*. Denote $T = \inf\{t : \frac{\partial \|\mathbf{w}(t)\|^2}{\partial t} \geq 0\}$. Under Assumption 1, we have*

$$\cos\theta(t) \geq \begin{cases} 1 - \dfrac{2}{e^{A_1 t + B_1} + 1}, & t \leq T, \\[2mm] 1 - \dfrac{2}{e^{A_2\sqrt{t-T+C_2}+B_2} + 1}, & t > T. \end{cases}$$

*Here $A_1 = \frac{2c_0}{\pi\|\mathbf{w}(0)\|} > 0, B_1 = -2\ln\left|\tan\frac{\theta(0)}{2}\right|, A_2 = \frac{4c_0}{\sqrt{0.6\pi}} > 0, B_2 = -2\ln\left|\tan\frac{\theta(T)}{2}\right| - \frac{4c_0\|\mathbf{w}(T)\|}{0.6\pi}, and C_2 = \frac{\|\mathbf{w}(T)\|^2}{0.6}.$*

This result shows that as $t \to \infty$, the error metric $1-\cos\theta(t)$ converges to 0, which implies $\mathbf{w}(t) \to \mathbf{v}$, i.e., the weight converges to the globally optimal direction. Additionally, the training goes through two stages: in the first stage, $1 - \cos\theta(t) \leq O(e^{-A_1 t})$; in the second stage, $1 - \cos\theta(t) \leq O(e^{-A_2\sqrt{t}})$.

Next, we present an interesting characterization of the optimization trajectory: the weight norm goes through a descending stage and an ascending stage. This "descending-ascending pattern" can be established for deep linear nets as well (see Theorem 3 later).

**Lemma 1** *Denote $T := \inf\{t : \frac{\partial \|\mathbf{w}(t)\|^2}{\partial t} \geq 0\}$. Under Assumption 1, we have $T < \infty$, and*

$$\frac{\partial \|\mathbf{w}(t)\|^2}{\partial t} \begin{cases} \geq 0, & \forall\, t \geq T. \\ < 0, & \forall\, t < T. \end{cases} \tag{3}$$

This lemma shows that the training dynamics can be divided into two stages: in the first stage, the norm $\|\mathbf{w}(t)\|$ is decreasing; in the second stage, the norm $\|\mathbf{w}(t)\|$ is increasing. The first stage might be non-existent if $\partial\|\mathbf{w}(0)\|^2/\partial t \geq 0$, and the second stage is always existent (i.e., $T < \infty$ as stated in the lemma). The existence of the second stage implies that there is a lower bound of the weight norm throughout training.

**Remark 1** *The lower bound of weight norm is not needed for establishing Theorem 1 (the convergence result in the logistic regression case), but will be crucial for multi-layer linear nets. The reason we present Lemma 1 is that its analysis is essentially similar to the multi-layer nets case while easier to understand.*

## 3.2 Basic properties for the proof of Theorem 1

In this subsection, we describe a few technical propositions that will be used in the proof of Theorem 1. Readers who care about the proof sketch can skip this part, and turn back to check these propositions if we mention subsequently.

**Proposition 1** *Denote $c := \mathbb{E}_{\mathbf{x} \sim \mathcal{D}} \|\mathbf{x}\|$. Suppose $\phi(\mathbf{x}, \mathbf{w}) = \mathbf{w}^\top \mathbf{x}$ and $P(\{\mathbf{x} : \mathbf{v}^\top \mathbf{x} = 0\}) = 0$. Then the function $L(\mathbf{w})$ defined in Eq. (1) is $c$-Lipschitz continuous, $\frac{1}{4}\lambda_{\max}(\Sigma)$-smooth, and convex but not strongly convex.*

Generally speaking, gradient methods do not exhibit linear convergence rate for non-strongly convex functions [40]. However, the training dynamic enjoy faster convergence rates in terms of directional convergence, due to certain invariant properties (such as Proposition 2 below).

**Proposition 2** *Suppose $\phi(\mathbf{x}, \mathbf{w}) = \mathbf{w}^\top \mathbf{x}$ and $P(\{\mathbf{x} : \mathbf{v}^\top \mathbf{x} = 0\}) < 1$.*

*Then the gradient flow for the problem in Eq. (1) satisfies the following properties: (i) $\mathbf{v}^\top \nabla L(\mathbf{w}) < 0$ for any $\mathbf{w} \in \mathbb{R}^d$. (ii) $\mathbf{v}^\top \mathbf{w}(t)$ is increasing as $t$ increases; (iii) $\{\|\mathbf{w}(t)\|\}_{t \geq 0}$ is unbounded;*

*Similarly, the gradient descent for the problem in Eq. (1) satisfies the following properties: (iv) if the learning rates are lower bounded, i.e., $\eta_n \geq \eta_- > 0$, then $\mathbf{v}^\top \mathbf{w}(n)$ is increasing as $n$ increases, and $\{\|\mathbf{w}(n)\|\}_{n=0}^\infty$ is unbounded.*

When Assumption 1 holds, we can provide a more precise description of $\mathbf{v}^\top \mathbf{w}(t)$ and $\nabla L(\mathbf{w})$ below:

**Proposition 3** *Under Assumption 1, we have: (i) $\|\nabla L(\mathbf{w})\| \leq c_0 = \mathbb{E}_{\mathbf{x} \in \mathcal{D}_2} \|\mathbf{x}\|$; (ii) $\nabla L(a\mathbf{v}) = \mathbb{E}_{t \sim \mathcal{D}_{\mathbf{v}}} \frac{|t|}{1 + e^{a|t|}} \mathbf{v}$; (iii) $\lim_{t \to \infty} \mathbf{v}^\top \mathbf{w}(t) = +\infty$. (iv) If the learning rates are lower bounded, i.e., $\eta_n \geq \eta_- > 0$, then $\lim_{n \to \infty} \mathbf{v}^\top \mathbf{w}(n) = +\infty$.*

**Remark 2** *This proposition shows that in the logistic regression case, we do not need to consider the case $\mathbf{w}(t) \parallel \mathbf{v}$ (including $\mathbf{0}$), since we have $\nabla L(\mathbf{w}(t)) \parallel \mathbf{v}$ when $\mathbf{w}(t) \parallel \mathbf{v}$ by (ii) in Proposition 3. Therefore, we only need to consider the case $\mathbf{w}(t) \neq \mathbf{0}$ and $\mathbf{w}(t) \nparallel \mathbf{v}, \forall t \geq 0$.*

## 3.3 Proof sketch of Theorem 1

We mainly consider the dynamics of $\cos\theta(\mathbf{w}, \mathbf{v})$, which is also employed in [29, 28, 50, 43].

$$\frac{\partial \cos\theta(t)}{\partial t} = -\underbrace{\frac{1}{\|\mathbf{w}(t)\|}}_{I_1} \cdot \underbrace{\left(\mathbf{v} - \left(\overline{\mathbf{w}}(t)^\top \mathbf{v}\right)\overline{\mathbf{w}}(t)\right)^\top \nabla L(\mathbf{w}(t))}_{I_2}. \tag{4}$$

**Step 1:** We have the analytic expression of $I_2$ shown in Lemma 2 below. This expression relies on Assumption 1, because the symmetry eliminates the collision among data points.

**Lemma 2** *Under Assumption 1 and if $\mathbf{w} \neq \mathbf{0}$, then*

$$-\left(\mathbf{v} - \left(\overline{\mathbf{w}}^\top \mathbf{v}\right)\overline{\mathbf{w}}\right)^\top \nabla L(\mathbf{w}) = -\mathbf{v}^\top \left(\mathbf{I}_d - \overline{\mathbf{w}}\,\overline{\mathbf{w}}^\top\right) \nabla L(\mathbf{w}) = \frac{c_0 \sin^2\theta(\mathbf{w}, \mathbf{v})}{\pi}. \tag{5}$$

*Moreover, if additionally $\mathbf{w} \nparallel \mathbf{v}$, the above result implies*

$$-\left(\mathbf{I}_d - \overline{\mathbf{w}}\,\overline{\mathbf{w}}^\top\right) \nabla L(\mathbf{w}) = \frac{c_0}{\pi}\left(\mathbf{I}_d - \overline{\mathbf{w}}\,\overline{\mathbf{w}}^\top\right)\mathbf{v}. \tag{6}$$

From Lemma 2 and Eq. (4), we obtain that $\frac{\partial \cos\theta(t)}{\partial t} > 0$ when $\mathbf{w}(t) \neq \mathbf{0}$ and $\mathbf{w}(t) \nparallel \mathbf{v}$.

**Step 2:** Next, we analyze the remaining term $\|\mathbf{w}(t)\|$ in Eq. (4) by its derivative. Denote

$$g(\mathbf{w}(t)) := \frac{1}{2} \cdot \frac{\partial \|\mathbf{w}(t)\|^2}{\partial t} = -\mathbf{w}(t)^\top \nabla L(\mathbf{w}(t)). \tag{7}$$

The following lemma describes the behavior of $g(\mathbf{w}(t))$. Some properties in Lemma 3 can be explained visually from Figure 2 in the Appendix.

**Lemma 3** *Under Assumption 1, the following hold:*

*(i) $g(\mathbf{w}) \leq 0.3$; (ii) If $\theta(\mathbf{w}, \mathbf{v}) > \pi/2$, then $g(\mathbf{w}) < 0$;*

*(iii) If $\theta(\mathbf{w}, \mathbf{v}) \leq \pi/2$, then $\partial g(\mathbf{w})/\partial \theta(\mathbf{w}, \mathbf{v}) < 0$.*

*(iv) If $0 < \|\mathbf{w}\| \leq \frac{c_0 |\cos \theta(\mathbf{w}, \mathbf{v})|}{2\pi L}$ with $L := \frac{1}{8}\mathbb{E}_{\mathbf{x} \sim \mathcal{D}_2} \|\mathbf{x}\|^2$, then $g(\mathbf{w}) \cos \theta(\mathbf{w}, \mathbf{v}) > 0$.*

**Step 3:** Finally, we bound $\|\mathbf{w}(t)\| \leq \|\mathbf{w}(0)\|$ if $t \leq T$, and $\|\mathbf{w}(t)\|^2 - \|\mathbf{w}(T)\|^2 \leq 0.6(t - T)$ if $t > T$ from (i) in Lemma 3, which shows two-phase directional convergence in Theorem 1.

### 3.4 Convergence of gradient descent for linear regression

Now we turn to the gradient descent setting based on the previous results. The difficulty we encounter is that the arbitrary choice of learning rates in each step may break the directional monotonicity in Lemma 2 as Eq. (8) reveals (See Appendix A for the proof).

$$\cos \theta(n+1) - \cos \theta(n) = \frac{1}{\|\mathbf{w}(n+1)\|} \left[ -\eta_n \left( \mathbf{v} - (\overline{\mathbf{w}}(n)^\top \mathbf{v})\overline{\mathbf{w}}(n) \right)^\top \nabla L(\mathbf{w}(n)) \right.$$
$$\left. - \left( \|\mathbf{w}(n+1)\| - \overline{\mathbf{w}}(n)^\top \mathbf{w}(n+1) \right) \cos \theta(n) \right]. \tag{8}$$

Fortunately, when $\mathbf{v}^\top \mathbf{w}(n) \leq 0$, we have $g(\mathbf{w}) \leq 0$ by (ii) in Lemma 3, implying that the first-phase directional convergence in the gradient flow case still holds.

**Lemma 4** *Under Assumption 1, suppose $\theta(0) \neq \pi$ and $\mathbf{v}^\top \mathbf{w}(0) < 0$. Then we have that*

$$\cos \theta(n) \geq 1 - (1 - \cos \theta(0)) e^{-BS_n^-} \tag{9}$$

*until $\cos \theta(n) \geq 0$, where $S_n^- := \sum_{k=0}^{n-1} \frac{\eta_k}{\sqrt{A + \sum_{i=0}^{k} \eta_i^2}}$, $A = \frac{\|\mathbf{w}(0)\|^2}{c_0^2}$ and $B = \frac{1 + \cos \theta(0)}{\pi}$.*

Hence, when $\mathbf{w}(n)$ stays in the "wrong" region that $\theta(n) > \pi/2$, larger learning rate gives faster directional convergence to the region $\{\mathbf{w} : \theta(\mathbf{w}, \mathbf{v}) \leq \pi/2\}$. Unfortunately, when $\theta(n) \leq \pi/2$, the directional dynamic becomes unstable and heavily relies on the current learning rate. However, after simple calculation invoked from Lemma 2, we find that (Lemma 12 in Appendix)

$$\|\mathbf{w}(n+1)\|^2 = \left(\overline{\mathbf{w}}(n)^\top \mathbf{w}(n+1)\right)^2 + \left(\frac{c_0 \eta_n}{\pi}\right)^2 \sin^2 \theta(n). \tag{10}$$

Combining Eq. (10) and Eq. (8), we figure out

$$\cos \theta(n+1) - \cos \theta(n) = \frac{1}{\|\mathbf{w}(n+1)\|} \left( \frac{c_0 \eta_n \sin^2 \theta(n)}{\pi} - \frac{c_0^2 \eta_n^2 \sin^2 \theta(n) \cos \theta(n)/\pi^2}{\|\mathbf{w}(n+1)\| + \overline{\mathbf{w}}_n^\top \mathbf{w}(n+1)} \right).$$

Hence, we characterize a sufficient condition in the remaining training period to guarantee the directional monotonicity. Moreover, we will show that such a condition can be satisfied when the weight norm $\|\mathbf{w}(n)\|$ is large enough compared to the current learning rate.

**Lemma 5 (A sufficient convergence condition)** *Under Assumption 1, if $\mathbf{w}(0) \neq \mathbf{0}$, $\theta(0) \neq \pi$, and there exists a $\delta > 0$, s.t.*

$$\|\mathbf{w}(n+1)\| + \overline{\mathbf{w}}(n)^\top \mathbf{w}(n+1) \geq \frac{(1 + \delta)c_0 \eta_n \cos \theta(n)}{\pi}, \forall\, n \in \mathbb{N}, \tag{11}$$

*then there exist constants $A, B, C > 0$ such that*

$$\cos \theta(n) \geq 1 - (1 - \cos \theta(0)) e^{-BS_n^+}. \tag{12}$$

*Here $S_n^+ := \sum_{k=0}^{n-1} \frac{\eta_k}{\sqrt{A + \sum_{i=0}^{k}(\eta_i^2 + C\eta_i)}}$, $A = \|\mathbf{w}(0)\|^2/c_0^2$, $B = \frac{\delta(1 + \cos \theta(0))}{(1+\delta)\pi}$, and $C = 0.6/c_0^2$.*

Furthermore, we can show the directional convergence with bounded learning rates as follows. Note that when $\|\mathbf{w}(n)\| \geq \eta_n c_0 + (1 + \delta)c_0 \eta_n/(2\pi)$, by (i) in Proposition 3, Eq. (11) can be satisfied from

$$\|\mathbf{w}(n+1)\| + \overline{\mathbf{w}}(n)^\top \mathbf{w}(n+1) \geq 2 \left( \|\mathbf{w}(n)\| - \eta_n \|\nabla L(\mathbf{w}(n))\| \right) \geq (1 + \delta)c_0 \eta_n/\pi.$$

Once $\eta_n \leq \eta_+$, then $\|\mathbf{w}(n)\| \geq R_1 := \eta_+ c_0 + c_0 \eta_+ / \pi$ is sufficient to derive the convergence. We note that $\|\mathbf{w}(n)\| \geq \mathbf{v}^\top \mathbf{w}(n)$, and from (ii) in Proposition 2 and (iii) in Proposition 3, the right term $\mathbf{v}^\top \mathbf{w}(n)$ monotonically increases to infinity. Thus, after finite iterations, the sufficient convergence condition would be satisfied.

**Theorem 2** *Under Assumption 1, for a bounded learning rate sequence $\{\eta_n\}_{n=1}^\infty$ with $\eta_+ \geq \eta_n \geq \eta_- > 0$, $\cos\theta(n)$ first converges following Eq. (9) until $\cos\theta(n_0) \geq 0$ for some $n_0 \geq 0$. Then there exists an $n_1 \geq n_0$ such that $\mathbf{v}^\top \mathbf{w}(n_1) \geq \eta_+ c_0 + \eta_+ c_0 / \pi$, and $\cos\theta(n)$ converges to 1 from $n_1$ following Eq. (12).*

**Remark 3** *Generally, a $\beta$-smooth objective loss has the learning rate constraint $(0 < \eta < \frac{2}{\beta})$ to guarantee convergence [40]. There is no such constraint because the purpose is learning the direction instead of decreasing the loss. Furthermore, we need to underline that the loss still converges. Since for large enough $\|\mathbf{w}(n)\|$, the smoothness coefficient of the objective becomes certainly small and the bounded learning rate is naturally suitable to guarantee convergence.*

**Comparison.** Our convergence bounds provide a faster and non-asymptotic directional convergence rate $\exp\left(\mathcal{O}(-\sqrt{t})\right)$ (under constant learning rates, i.e., $\eta_n = \Theta(1)$) compared to $\mathcal{O}\left(1/\log^2 t\right)$ in Soudry et al. [46, Theorem 5]. Additionally, previous results hold for a certain large $t$ and the finite dataset with a positive margin, but we show the behavior in the entire training dynamic. We think that the possible explanation is the benefit of the structured data in Assumption 1. Moreover, the prior technique of proofs mostly employs the decomposition: $\mathbf{w}(t) = \hat{\mathbf{w}} \log t + \boldsymbol{\rho}(t)$ with the max-margin solution $\hat{\mathbf{w}}$ and almost bounded residual term $\boldsymbol{\rho}(t)$. We can implicitly obtain an analogous decomposition, but such a way would lose sight of variation during the early training, such as the decreasing and increasing period of $\|\mathbf{w}(t)\|$. Furthermore, for gradient descent, directional convergence in previous results is built on the loss convergence [22, 24, 38, 39, 46]. Hence they need "small" learning rates with a data-related upper bound. When data are distributed well, we derive the directional convergence directly under implicitly bounded learning rates and then obtain the loss convergence from the directional convergence.

# 4 Deep Linear Networks

In this section, we extend the results of gradient flow to deep linear networks. For an $N$-layer linear network $\phi(\mathbf{x}, \mathbf{w}) = W_N \ldots W_1 \mathbf{x}$ where $\mathbf{w} := (W_N, \ldots, W_1)$, the objective is

$$\min_{\mathbf{w}} L^{(N)}(W_N, \ldots, W_1) := \mathbb{E}_{\mathbf{x} \sim \mathcal{D}} \ln\left[1 + e^{-y(\mathbf{x})W_N \cdots W_1 \mathbf{x}}\right].$$

Every such a network represents a linear mapping given by $\mathbf{w}_e = (W_N \cdots W_1)^\top \in \mathbb{R}^d$:

$$L^{(N)}(W_1, \ldots, W_N) = L^{(1)}(\mathbf{w}_e) = \mathbb{E}_{\mathbf{x} \sim \mathcal{D}} \ln\left(1 + e^{-y(\mathbf{x})\mathbf{w}_e^\top \mathbf{x}}\right).$$

## 4.1 Main results for deep linear networks

A key tool for analyzing the induced flow for $\mathbf{w}_e$ is established in Claim 2 of Arora et al. [3]. Specifically, if the initial balancedness conditions

$$W_{j+1}(0)^\top W_{j+1}(0) = W_j(0)^\top W_j(0), j = 1, \ldots, N-1 \tag{13}$$

hold, then we have the induced gradient flow with $\mathbf{w}_e(t)$:

$$\frac{\partial \mathbf{w}_e(t)}{\partial t} = -\|\mathbf{w}_e(t)\|^{2-\frac{2}{N}} \left(\frac{dL^{(1)}(\mathbf{w}_e(t))}{d\mathbf{w}} + (N-1)\overline{\mathbf{w}}_e(t)\overline{\mathbf{w}}_e(t)^\top \frac{dL^{(1)}(\mathbf{w}_e(t))}{d\mathbf{w}}\right). \tag{14}$$

Such an initialization technique is common for linear networks [2, 3, 7, 18, 19]. Claim 4 in Arora et al. [2] discusses the importance of (nearly) balanced initialization for deep linear networks. For equal-width linear nets, orthogonal intialization is a special case of balanced initialization, and was shown to perform better than Gaussian initialization [19].

Before presenting results of deep linear networks for general $N$, we first present the result for the simple case $N = 2$:

**Proposition 4** *Under Assumption 1 and the initial balancedness condition Eq. (13), if $N = 2$, $\mathbf{w}_e(0) \neq \mathbf{0}$ and $\theta(0) \neq \pi$, we obtain the directional convergence as follows:*

$$\cos\theta(t) = 1 - \frac{2}{C_1 e^{2c_0 t/\pi} + 1}, \quad \text{with } C_1 = \frac{1 + \cos\theta(0)}{1 - \cos\theta(0)}.$$

Next, we present a lemma which is an extension of Lemma 1 to the deep linear case: both lemmas present the "descending-then-ascending" pattern of the weight norm.

**Lemma 6** *Denote $T := \inf\{t : \partial\|\mathbf{w}_e(t)\|^2/\partial t \geq 0\}$. We have $T < \infty$, and*

$$\frac{\partial\|\mathbf{w}_e(t)\|^2}{\partial t} \begin{cases} \geq 0, & \forall\, t \geq T \\ < 0, & \forall\, t < T. \end{cases} \tag{15}$$

The main result for the deep linear net case provides the convergence rate of the gradient flow.

**Theorem 3** *Under Assumption 1 and the initial balancedness condition Eq. (13), if $N > 2$, $\mathbf{w}_e(0) \neq \mathbf{0}$ and $\theta(0) \neq \pi$, we obtain two-phase convergence below. With the $T$ defined in Lemma 6, we have*

$$\cos\theta(t) \geq \begin{cases} 1 - \dfrac{2}{C_1(A_1 t/B_1 + 1)^\alpha + 1}, & t \leq T, \\[2mm] 1 - \dfrac{2}{e^{A_2(t-T)+B_2} + 1}, & t > T. \end{cases}$$

*Here $A_1 = (N-2)c_0$, $B_1 = \|\mathbf{w}_e(0)\|^{\frac{2}{N}-1}$, $C_1 = \frac{1+\cos\theta(0)}{1-\cos\theta(0)}$, and $A_2 = 2c_0\|\mathbf{w}_e(T)\|^{1-\frac{2}{N}}/\pi$, $B_2 = -2\ln\left|\tan\frac{\theta(T)}{2}\right|$, and $\alpha = 2/[(N-2)\pi]$.*

*In addition, we have the upper bound that*

$$\cos\theta(t) \leq 1 - \frac{2}{e^{F[(0.6t+D)^{N/2} - D^{N/2}]+E} + 1},$$

*where $D = \|\mathbf{w}_e(0)\|^{\frac{2}{N}}$, $E = -2\ln\left|\frac{\tan\theta(0)}{2}\right|$, and $F = \frac{4c_0}{0.6N\pi}$.*

Theorem 3 implies $\cos\theta(t)$ undergoes two stages: polynomial convergence stage and exponential convergence stage. Finally, as for the initialization $\theta(0) = \pi$, we have that $\theta(t) = \pi$, $\forall t \geq 0$, and $\mathbf{w}_e(t) \to \mathbf{0}$ but never hits the origin by the lower bound in Eq. (17).

**Remark 4** *We briefly discuss how different items affect the convergence rate of both stages. The convergence speed of the second stage is positively correlated with $A_2$, which is positively correlated with $c_0$ and $\|\mathbf{w}_e(T)\|^{1-2/N}$ (when the number of layers $N \geq 3$). Scaling up the data can increase the expected data norm $c_0$, thus increasing $A_2$ and the convergence speed. It is not clear, though, how to increase $\|\mathbf{w}_e(T)\|$ since it depends on an unknown transition time $T$. The convergence speed of the first stage is positively correlated with $A_1$ and $1/B_1$. We skip the discussions of this stage.*

**Remark 5** *Since we need to cover the worse case during optimization, our bound may seem loose in two possible cases. First, when $t \leq T$, we use the lower bound in Eq. (17) to capture the decreasing period of $\|\mathbf{w}_e(t)\|$. Actually, $\|\mathbf{w}_e(t)\|$ may descend to a certain value that is higer than the lower bound in Eq. (17) before ascending. Second, when $t \geq T$, note that $\cos\theta(t) \to 1$ except $\mathbf{w}_e(0) = k\mathbf{v}$ for $k \leq 0$ by Theorem 3. We can guarantee $\|\mathbf{w}_e(t)\| \to \infty$ because $\mathbf{v}^\top \mathbf{w}_e(t)$ is increasing after some time (but not always). Hence, the convergence could be faster when $\|\mathbf{w}_e(t)\|$ increases much, but we only treat such a scheme lower bounded by $\|\mathbf{w}_e(T)\|$.*

**Comparison.** Our global convergence rate results for deep linear networks neither require small training loss [37, 21, 23] nor presume convergence of loss and weight direction [17, 38]. Moreover, our results do not require overparameterization [9, 11, 19]. However, we need the exact initial balancedness condition [3]. Previous works [37, 21, 51] mainly provided asymptotic directional convergence to the maximum margin solution given zero classification error at the beginning. We could obtain an explicit global directional convergence including the early training dynamic. Our directional rates could be linear when the induced weight norm begins increasing, which is faster than the polynomial convergence rate of the surrogate loss function obtained in [37, Theorem 4.3]. However, our results heavily depend on the benign training data assumption and the linear net structure compared to Lyu and Li [37], Yun et al. [51].

## 4.2 Technical lemmas for analyzing deep linear networks

We present a series of technical results that will be used for proving Lemma 6 and Theorem 3. Some are extensions of those for logstic regression, and some are new for deep nets.

**Lemma 7** *Under Assumption 1 and the initial balancedness condition Eq. (13), if $\mathbf{w}_e(t) \neq \mathbf{0}$, then*

$$\frac{\partial \cos \theta(\mathbf{w}_e(t), \mathbf{v})}{\partial t} = \frac{c_0 \sin^2 \theta(\mathbf{w}_e(t), \mathbf{v})}{\pi} \cdot \|\mathbf{w}_e(t)\|^{1-\frac{2}{N}} . \tag{16}$$

The main difference from the logistic regression case is that the dependence of weight norm $\|\mathbf{w}_e(t)\|$ is reversed. Larger $\|\mathbf{w}_e(t)\|$ gives faster convergence for the deep linear networks when $N \geq 3$, while no influence for $N = 2$, and for $N = 1$ it is opposite (i.e. larger $\|\mathbf{w}_e(t)\|$ gives slower convergence).

We then provide a few properties of the weight norm during training.

**Lemma 8** *Under Assumption 1 and the initial balancedness condition Eq. (13) with $\mathbf{w}_e(0) \neq \mathbf{0}$, we have the following two properties:*

*(i) Suppose $N > 2$. For any $t \geq 0$, we have*

$$\left( \|\mathbf{w}_e(0)\|^{\frac{2}{N}} + 0.6t \right)^{\frac{N}{2}} \geq \|\mathbf{w}_e(t)\| \geq \left( \|\mathbf{w}_e(0)\|^{\frac{2}{N}-1} + (N-2)c_0 t \right)^{-\frac{N}{N-2}} > 0. \tag{17}$$

*(ii) Suppose $\partial \|\mathbf{w}_e(t_0)\|^2 / \partial t = 0$ with $\mathbf{w}_e(t_0) \neq \mathbf{0}$ for some $t_0 \geq 0$. Then $\partial g(\mathbf{w}_e(t_0)) / \partial t > 0$.*

We briefly explain how to use Lemma 8 to prove Lemma 6, which consists of two properties: Eq. (15) and $T < \infty$. First, Eq. (15) is an immediate consequence of Part (ii) of Lemma 8, as well as a calculus fact Lemma 10 (see Appendix). This proof approach is similar to that for the logistic regression case. Second, $T < \infty$ requires a different proof approach from the logistic regression case: the latter is due to unbounded weight norm and the fact that zero-norm weight is non-stationary, but for deep linear case the latter does not hold (the origin is a saddle point). Instead, we rely on Part (i) of Lemma 8 to prove $T < \infty$ by contradiction. More specifically, assume $T = \infty$, i.e., $\|\mathbf{w}_e(t)\|$ is decreasing for all $t$, then $\|\mathbf{w}_e(t)\|$ converges to a finite value. We will use Part (i) of Lemma 8 and Part (iv) of Lemma 3 to derive a contradiction. See details in Appendix B.2.

## 4.3 Proof of Theorem 3

Proof: Using Lemma 7 and the lower bound of $\|\mathbf{w}_e(t)\|$ in Lemma 8, we obtain

$$\frac{1}{2} \ln \frac{1 + \cos \theta(t)}{1 - \cos \theta(t)} - \frac{1}{2} \ln \frac{1 + \cos \theta(0)}{1 - \cos \theta(0)} \geq \frac{1}{(N-2)\pi} \ln \frac{A_1 t + B_1}{B_1}.$$

This implies

$$\cos \theta(t) \geq 1 - \frac{2}{C_1 (A_1 t / B_1 + 1)^\alpha + 1}, \forall t \geq 0. \tag{18}$$

Notice that (18) holds for the whole training period, thus surely holds for $t \leq T$.

Next, we will show a stronger convergence rate for the peoriod $t > T$, i.e., the second part of the theorem. According to Lemma 6 we have

$$\frac{\partial \|\mathbf{w}_e(t)\|^{\frac{2}{N}}}{\partial t} \geq 0, \ \forall t \geq T.$$

Hence $\|\mathbf{w}_e(t)\| \geq \|\mathbf{w}_e(T)\|, \forall t \geq T$. Therefore,

$$\frac{1}{2} \ln \frac{1 + \cos \theta(t)}{1 - \cos \theta(t)} - \frac{1}{2} \ln \frac{1 + \cos \theta(T)}{1 - \cos \theta(T)} \geq \frac{c_0}{\pi} \|\mathbf{w}_e(T)\|^{1-\frac{2}{N}} (t - T).$$

$$\implies \quad \cos \theta(t) \geq 1 - \frac{2}{e^{A_2(t-T)+B_2} + 1}, \forall \, t \geq T.$$

For the third part, using the upper bound of $\|\mathbf{w}_e(t)\|$ in Lemma 8, we obtain

$$\frac{1}{2} \ln \frac{1 + \cos \theta(t)}{1 - \cos \theta(t)} - \frac{1}{2} \ln \frac{1 + \cos \theta(0)}{1 - \cos \theta(0)} \leq \frac{2c_0}{0.6N\pi} \left( (0.6t + D)^{\frac{N}{2}} - D^{\frac{N}{2}} \right).$$

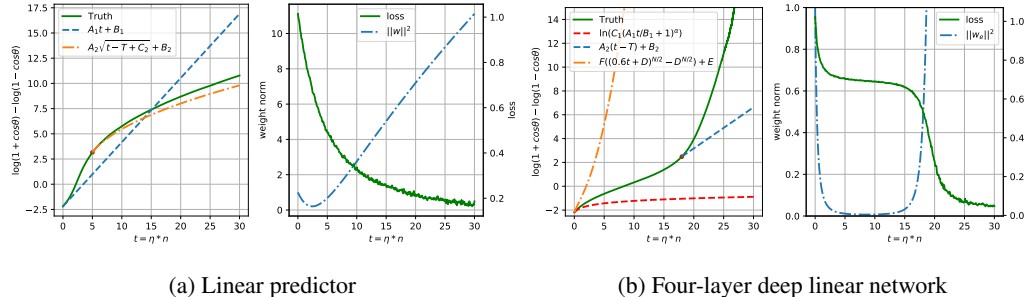

|   | (a) Linear predictor | (b) Four-layer deep linear network |
|---|---|---|

Figure 1: Simulation of (deep) linear network with $\mathbf{x} \sim \mathcal{U}(\mathcal{S}^1)$, $\mathbf{v} = (0,1)^\top$. (a): Linear predictor. We show our lower bounds in Theorem 1 at $n = 0$ and $n = 5000$. (b): Four-layer deep linear network. We also show our lower and upper bounds in Theorem 3 at $n = 0$ and $n = 18000$. In each experiment, we plot angle variation, loss and weight norm in sequence.

This implies

$$\cos\theta(t) \leq 1 - \frac{2}{e^{F[(0.6t+D)^{N/2} - D^{N/2}] + E} + 1}.$$

$\square$

## 5  Experiments

In this section we conduct experiments to verify our theoretical analyses. We consider training a linear classifier and a 4-layer linear network under the logit loss as our theorems show. We construct simple dataset with $\mathbf{x} \sim \mathcal{U}(\mathcal{S}^1)$ and $y(\mathbf{x}) = \text{sgn}(\mathbf{v}^\top \mathbf{x})$ with $\mathbf{v} = (0,1)^\top$. We use common stochastic gradient descent (SGD) with the batch size $1000$ and the constant small learning rate $10^{-3}$. Moreover, we choose an initial value $\mathbf{w}(0) = \mathbf{w}_e(0) = (0.6, -0.8)^\top$. In the deep linear network, we set $W_N(0) = \boldsymbol{u}_N^\top$, $W_i(0) = \boldsymbol{u}_{i+1}\boldsymbol{u}_i^\top$ with $\|\boldsymbol{u}_i\| = 1, i = 2, \ldots, N$ and $\boldsymbol{u}_1 = \mathbf{w}_e(0)$ to satisfy the balancedness conditions Eq. (13). The results are shown in Figure 1.

Figure 1a shows the optimization period for linear classifiers. We also plot the convergence bounds obtained in Theorems 1. Although we do not give convergence for SGD, our bounds in gradient flow still roughly match the directional convergence in practice, and the weight norm $\|\mathbf{w}(n)\|$ indeed goes through a decreasing and increasing period. Figure 1b depicts the dynamic for the deep linear networks. We also plot three convergence bounds in Theorems 3, which roughly match the actual behavior. The weight norm $\|\mathbf{w}_e(n)\|$ still goes through the decreasing and increasing period, and we could observe a distinct stuck period when $5 \leq \eta \cdot n \leq 10$ as the lower bound shown in Lemma 8. Moreover, the loss decreases slowly when $\|\mathbf{w}(n)\|$ is small, but $\cos\theta(t)$ still has considerable growth. Thus previous loss-based analysis may neglect potential variation of the accuracy we most concern. Furthermore, we can see a more rigorous understanding of $\|\mathbf{w}(n)\|$ may give more precise rates in the late training period ($t \geq 20$).

## 6  Conclusion

In this work, we have studied the training dynamic of (deep) linear networks in binary classification tasks. Specifically, we focus on the gradient flow and gradient descent methods under population loss. We have proved exact directional convergence for (deep) linear networks in the entire training process on spherically symmetric data. Moreover, we have characterized the descent and ascent behavior of the induced weight norm theoretically. Such a phenomenon also is observed in numerical experiments. We hope that our limited view of directional convergence would bring a better understanding of (linear) neural networks.

## Acknowledgments and Disclosure of Funding

Lin and Zhang have been supported by the National Key Research and Development Project of China (No. 2018AAA0101004) and Beijing Natural Science Foundation (Z190001).

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
