# A Missing Proofs in Section 3

## A.1 Proof of Proposition 1

Proof: $L(\mathbf{w})$ is Lipschitz continuous following from

$$\|\nabla L(\mathbf{w})\| \overset{(2)}{\leq} \mathbb{E}_{\mathbf{x}\sim\mathcal{D}} \left\| \frac{y(\mathbf{x})}{1 + e^{y(\mathbf{x})\mathbf{w}^\top\mathbf{x}}} \mathbf{x} \right\| \leq \mathbb{E}_{\mathbf{x}\sim\mathcal{D}} \|\mathbf{x}\| \leq \sqrt{\mathbb{E}_{\mathbf{x}\sim\mathcal{D}} \|\mathbf{x}\|^2} = \sqrt{tr(\Sigma)}.$$

The convexity of $L(\mathbf{w})$ is proved by

$$\nabla^2 L(\mathbf{w}) = \mathbb{E}_{\mathbf{x}\sim\mathcal{D}} \frac{1}{1 + e^{y(\mathbf{x})\mathbf{w}^\top\mathbf{x}}} \cdot \frac{e^{y(\mathbf{x})\mathbf{w}^\top\mathbf{x}}}{1 + e^{y(\mathbf{x})\mathbf{w}^\top\mathbf{x}}} \mathbf{x}\mathbf{x}^\top \succeq \mathbf{0}.$$

Moreover, for any $\mathbf{u} \in \mathbb{R}^d$ with $\|\mathbf{u}\| = 1$,

$$\mathbf{u}^\top \nabla^2 L(\mathbf{w})\mathbf{u} = \mathbb{E}_{\mathbf{x}\sim\mathcal{D}} \frac{1}{1 + e^{y(\mathbf{x})\mathbf{w}^\top\mathbf{x}}} \frac{e^{y(\mathbf{x})\mathbf{w}^\top\mathbf{x}}}{1 + e^{y(\mathbf{x})\mathbf{w}^\top\mathbf{x}}} \|\mathbf{x}^\top\mathbf{u}\|^2 \leq \frac{1}{4} \mathbb{E}\|\mathbf{x}^\top\mathbf{u}\|^2 \leq \frac{1}{4}\lambda_{\max}(\Sigma).$$

However, note that

$$0 \leq \frac{1}{1 + e^{k|\mathbf{v}^\top\mathbf{x}|}} \cdot \frac{e^{k|\mathbf{v}^\top\mathbf{x}|}}{1 + e^{k|\mathbf{v}^\top\mathbf{x}|}} \|\mathbf{x}\|^2 \leq \frac{\|\mathbf{x}\|^2}{4},$$

then from dominated convergence theorem, we have

$$\lim_{k\to+\infty} \mathbb{E}_{\mathbf{x}\sim\mathcal{D}} \frac{1}{1 + e^{k|\mathbf{v}^\top\mathbf{x}|}} \cdot \frac{e^{k|\mathbf{v}^\top\mathbf{x}|}}{1 + e^{k|\mathbf{v}^\top\mathbf{x}|}} \|\mathbf{x}\|^2 = \mathbb{E}_{\mathbf{x}\sim\mathcal{D}} \lim_{k\to+\infty} \frac{1}{1 + e^{k|\mathbf{v}^\top\mathbf{x}|}} \cdot \frac{e^{k|\mathbf{v}^\top\mathbf{x}|}}{1 + e^{k|\mathbf{v}^\top\mathbf{x}|}} \|\mathbf{x}\|^2$$
$$= \mathbb{E}_{\mathbf{x}\sim\mathcal{D}}\|\mathbf{x}\|^2 \mathbb{1}_{\{\mathbf{v}^\top\mathbf{x}=0\}} = 0, \tag{19}$$

where the final equality uses the assumption $P(\{\mathbf{x} : \mathbf{v}^\top\mathbf{x} = 0\}) = 0$. Then we get

$$\limsup_{k\to+\infty, \mathbf{w}=k\mathbf{v}} \|\nabla^2 L(\mathbf{w})\|_F \leq \limsup_{k\to+\infty} \mathbb{E}_{\mathbf{x}\sim\mathcal{D}} \frac{1}{1 + e^{k|\mathbf{v}^\top\mathbf{x}|}} \cdot \frac{e^{k|\mathbf{v}^\top\mathbf{x}|}}{1 + e^{k|\mathbf{v}^\top\mathbf{x}|}} \|\mathbf{x}\|^2 \overset{(19)}{=} 0.$$

Therefore

$$\lim_{\substack{\mathbf{w}=k\mathbf{v} \\ k\to+\infty}} \nabla^2 L(\mathbf{w}) = \mathbf{0}.$$

Thus $L(\mathbf{w})$ is not strongly convex. $\qquad\square$

## A.2 Proof of Proposition 2

Proof: Proof of (i): Suppose $\|\mathbf{w}\| \leq M$ for some positive constant $M$, and notice that

$$-\mathbf{v}^\top \nabla L(\mathbf{w}) = \mathbb{E}_{\mathbf{x}\sim\mathcal{D}} \frac{|\mathbf{v}^\top\mathbf{x}|}{1 + e^{y(\mathbf{x})\mathbf{w}^\top\mathbf{x}}} \geq 0.$$

More specifically,

$$-\mathbf{v}^\top \nabla L(\mathbf{w}) \geq \mathbb{E}_{\mathbf{x}\sim\mathcal{D}} \frac{|\mathbf{v}^\top\mathbf{x}|}{1 + e^{\|\mathbf{w}\|\cdot\|\mathbf{x}\|}} \geq \frac{\mathbb{E}_{\mathbf{x}\sim\mathcal{D}}|\mathbf{v}^\top\mathbf{x}|\mathbb{1}_{\{\|\mathbf{x}\|\leq R\}}}{1 + e^{MR}} \geq \frac{r \cdot P\left(\|\mathbf{x}\| \leq R, |\mathbf{v}^\top\mathbf{x}| \geq r\right)}{1 + e^{MR}}.$$

Let $\epsilon := P\left(|\mathbf{v}^\top\mathbf{x}| > 0\right)$, then $\epsilon > 0$ from assumption $P(\{\mathbf{x} : \mathbf{v}^\top\mathbf{x} = 0\}) < 1$. There exist $R > r > 0$, such that $P\left(\|\mathbf{x}\| \leq R, |\mathbf{v}^\top\mathbf{x}| \geq r\right) \geq \epsilon/2$, then $-\mathbf{v}^\top\nabla L(\mathbf{w}(t)) \geq 0.5r\epsilon/(1 + e^{MR}) > 0$.

Proof of (ii): According to (i), we have $\partial(\mathbf{v}^\top\mathbf{w}(t))/\partial t > 0$ and $\mathbf{v}^\top(\mathbf{w}(n+1) - \mathbf{w}(n)) > 0$, showing that $\mathbf{v}^\top\mathbf{w}(t)$ and $\mathbf{v}^\top\mathbf{w}(n)$ are increasing.

Proof of (iii): Assume the contrary that $\|\mathbf{w}(t)\| \leq M, \forall t \geq 0$ for certain $M$. Then $\mathbf{v}^\top\mathbf{w}(t) \leq \|\mathbf{w}(t)\| \leq M, \forall t \geq 0$. Hence, $\mathbf{v}^\top\mathbf{w}(t)$ converges from (ii), showing that $\lim_{t\to\infty} \mathbf{v}^\top\nabla L(\mathbf{w}(t)) = 0$. Using the previous argument again, we obtain a contradiction. Thus the assumption does not hold, i.e., $\{\|\mathbf{w}(t)\|\}_{t\geq 0}$ is unbounded.

Proof of (iv): Assume the contrary that $\|\mathbf{w}(n)\| \leq M, \forall n \geq 0$ for certain $M$. Then $\mathbf{v}^\top\mathbf{w}(n) \leq \|\mathbf{w}(n)\| \leq M, \forall n \geq 0$. Hence, $\mathbf{v}^\top\mathbf{w}(n)$ converges from (ii), showing that $\lim_{n\to\infty} \eta_n \mathbf{v}^\top\nabla L(\mathbf{w}(n)) = 0$. Since $\eta_n \geq \eta_- > 0$, then $\lim_{n\to\infty} \mathbf{v}^\top\nabla L(\mathbf{w}(n)) = 0$, remaining the same argument as the gradient flow case. $\qquad\square$

## A.3 Proof of Proposition 3

**Proposition 5 (Restatement of Proposition 3.)** *Under Assumption 1, we have: (i)* $\|\nabla L(\mathbf{w})\| \leq c_0 = \mathbb{E}_{\mathbf{x} \in \mathcal{D}_2} \|\mathbf{x}\|$; *(ii)* $\nabla L(a\mathbf{v}) = \mathbb{E}_{t \sim \mathcal{D}_\mathbf{v}} \frac{|t|}{1 + e^{a|t|}} \mathbf{v}$; *(iii)* $\lim_{t \to \infty} \mathbf{v}^\top \mathbf{w}(t) = +\infty$. *(iv) If the learning rates are lower bounded, i.e.,* $\eta_n \geq \eta_- > 0$*, then* $\lim_{n \to \infty} \mathbf{v}^\top \mathbf{w}(n) = +\infty$.

Proof: Recall that $\mathcal{D}_{\mathbf{w},\mathbf{v}}$ denotes the marginal distribution of $\mathbf{x} = (x_1, \ldots, x_d)^\top \sim \mathcal{D}$ on the subspace spanned by two linearly independent vectors $\mathbf{w}, \mathbf{v}$ (a distribution over $\mathbb{R}^2$),

Since $\mathcal{D}$ is spherically symmetric, and $\frac{y(\mathbf{x})}{1 + e^{y(x)\mathbf{w}^\top \mathbf{x}}} = \frac{\text{sgn}(\mathbf{x}^T \mathbf{v})}{1 + e^{\text{sgn}(\mathbf{x}^T \mathbf{v})\mathbf{w}^\top \mathbf{x}}}$ is only decided by $\mathbf{v}^\top \mathbf{x}$ and $\mathbf{w}^\top \mathbf{x}$, we could marginalize the distribution to $\text{span}\{\mathbf{v}, \mathbf{w}\}$, leading to

$$\nabla L(\mathbf{w}) = \mathbb{E}_{\mathbf{x} \sim \mathcal{D}} \frac{y(\mathbf{x})\mathbf{x}}{1 + e^{y(\mathbf{x})\mathbf{w}^\top \mathbf{x}}} = \mathbb{E}_{\mathbf{x} \sim \mathcal{D}_{\mathbf{w},\mathbf{v}}} \frac{y(\mathbf{x})\mathbf{x}}{1 + e^{y(\mathbf{x})\mathbf{w}^\top \mathbf{x}}} \in \text{span}(\mathbf{w}, \mathbf{v}). \quad (20)$$

Proof of (i):

$$\|\nabla L(\mathbf{w})\| \stackrel{(20)}{=} \left\| \mathbb{E}_{\mathbf{x} \sim \mathcal{D}_{\mathbf{w},\mathbf{v}}} \frac{y(\mathbf{x})\mathbf{x}}{1 + e^{y(\mathbf{x})\mathbf{w}^\top \mathbf{x}}} \right\| \leq \mathbb{E}_{\mathbf{x} \sim \mathcal{D}_{\mathbf{w},\mathbf{v}}} \|\mathbf{x}\| = \mathbb{E}_{\mathbf{x} \sim \mathcal{D}_2} \|\mathbf{x}\| = c_0.$$

The second equality uses the property that $\mathcal{D}$ is spherically symmetric, then $\mathcal{D}_{\mathbf{w},\mathbf{v}}$ is identical distribution for any $\mathbf{w}, \mathbf{v}$. We choose the first two coordinate directions for example.

Proof of (ii): when $\mathbf{w} = a\mathbf{v}$, $\mathcal{D}_{\mathbf{w},\mathbf{v}}$ reduce to $\mathcal{D}_\mathbf{v}$. Thus, we only need to consider the expectation on $\{\mathbf{x} : \mathbf{x} = t\mathbf{v}, t \in \mathbb{R}\}$, which leads to $y(\mathbf{x})\mathbf{x} = \text{sgn}(\mathbf{v}^\top \mathbf{x})\mathbf{x} = |t|\mathbf{v}$, and

$$\nabla L(a\mathbf{v}) \stackrel{(20)}{=} \mathbb{E}_{\mathbf{x} \sim \mathcal{D}_{\mathbf{w},\mathbf{v}}} \frac{y(\mathbf{x})\mathbf{x}}{1 + e^{y(\mathbf{x})a\mathbf{v}^\top \mathbf{x}}} = \mathbb{E}_{t \sim \mathcal{D}_\mathbf{v}} \frac{|t|}{1 + e^{a|t|}} \mathbf{v}.$$

Proof of (iii) and (iv): Finally, letting $\mathbf{w} = \alpha\mathbf{v} + \beta\mathbf{z}$, where $\mathbf{z} \perp \mathbf{v}$ and $\|\mathbf{z}\| = 1$, by dominated convergence theorem, we have

$$\lim_{\beta \to +\infty} -\mathbf{v}^\top \nabla L(\mathbf{w}) = \mathbb{E}_{\mathbf{x} \sim \mathcal{D}} \lim_{\beta \to +\infty} \frac{|\mathbf{v}^\top \mathbf{x}|}{1 + e^{\beta y(\mathbf{x})\mathbf{z}^\top \mathbf{x} + \alpha|\mathbf{v}^\top \mathbf{x}|}} = \mathbb{E}_{(\mathbf{z}^\top \mathbf{x}) \cdot (\mathbf{v}^\top \mathbf{x}) \leq 0} |\mathbf{v}^\top \mathbf{x}| = \frac{c_0}{\pi}, \quad (21)$$

where the last equality uses the distribution assumption.

Now we denote $\mathbf{w}(t) = \alpha(t)\mathbf{v} + \beta(t)\mathbf{z}(t)$. Assume the contrary, that $\mathbf{v}^\top \mathbf{w}(t) = \alpha(t)$ is bounded. Then from (iii) in Proposition 2, $\|\mathbf{w}(t)\|$ is unbounded, which implies $\beta(t)$ is unbounded. Since we have assumed $\alpha(t)$ is bounded, and (ii) in Proposition 2 implies $\alpha(t)$ is increasing, we obtain that $\alpha(t)$ converges to a certain finite value. This further implies that $-\mathbf{v}^\top \nabla L(\mathbf{w}(t)) \to 0$, which is a contradiction to (21). Thus the earlier assumption does not hold, i.e., $\lim_{t \to \infty} \mathbf{v}^\top \mathbf{w}(t) = +\infty$.

The same argument holds for $\mathbf{w}(n)$ if the learning rates are lower bounded. $\qquad \square$

## A.4 Proof of Lemma 2

Proof: As mentioned earlier, we have

$$\nabla L(\mathbf{w}) = -\mathbb{E}_{\mathbf{x} \sim \mathcal{D}} \frac{y(\mathbf{x})\mathbf{x}}{1 + e^{y(\mathbf{x})\mathbf{w}^\top \mathbf{x}}} \stackrel{(20)}{=} -\mathbb{E}_{\mathbf{x} \sim \mathcal{D}_{\mathbf{w},\mathbf{v}}} \frac{y(\mathbf{x})\mathbf{x}}{1 + e^{y(\mathbf{x})\mathbf{w}^\top \mathbf{x}}} \in \text{span}\{\mathbf{w}, \mathbf{v}\}.$$

Thus we only need to restrict our discussion to the two-dimensional linear space $\text{span}\{\mathbf{w}, \mathbf{v}\}$. With a little abuse of notation, we assume $\mathbf{w}, \mathbf{v} \in \mathbb{R}^2$ from now and use $\mathbf{w}, \mathbf{v}$ them as the representations of $\mathbf{w}, \mathbf{v}$ in the subspace $\text{span}\{\mathbf{w}, \mathbf{v}\}$ [2]. Similarly, we assume $\mathbf{x} \in \mathbb{R}^2$ from now on as well, and assume $\mathcal{D}_{\mathbf{w},\mathbf{v}}$ is a distribution on $\mathbb{R}^2$ (corresponding to the original distribution on $\text{span}\{\mathbf{w}, \mathbf{v}\}$). Then we have

$$\rho := -\left(\mathbf{v} - (\overline{\mathbf{w}}^\top \mathbf{v})\overline{\mathbf{w}}\right)^\top \nabla L(\mathbf{w}) = \mathbb{E}_{\mathbf{x} \sim \mathcal{D}_{\mathbf{w},\mathbf{v}}} \frac{|\mathbf{v}^\top \mathbf{x}| - \text{sgn}(\mathbf{v}^\top \mathbf{x})(\overline{\mathbf{w}}^\top \mathbf{v})(\overline{\mathbf{w}}^\top \mathbf{x})}{1 + e^{\text{sgn}(\mathbf{v}^\top \mathbf{x})\mathbf{w}^\top \mathbf{x}}}. \quad (22)$$

---

[2] Rigorously speaking, we would need to use different notations such as $\hat{\mathbf{w}} \in \mathbb{R}^2$ to represent the coordinate of $\mathbf{w}$ in $\text{span}\{\mathbf{w}, \mathbf{v}\}$. Anyhow, for simplicity we just reuse the notation $\mathbf{w}, \mathbf{v}$.

Additionally, the expression above is invariant to the rotation of the coordinate frame on $\text{span}\{\mathbf{w}, \mathbf{v}\}$ due to the spherical symmetry of $\mathcal{D}_{\mathbf{w},\mathbf{v}}$, so we can assume without loss of generality that $\overline{\mathbf{w}} = (1,0)^\top$, $\mathbf{v} = (v_1, v_2)^\top$. Then the numerator of (22) (for a given $\mathbf{x}$ inside the expectation) can be simplified to

$$|\mathbf{v}^\top \mathbf{x}| - \text{sgn}(\mathbf{v}^\top \mathbf{x})\left(\overline{\mathbf{w}}^\top \mathbf{v}\right)\left(\overline{\mathbf{w}}^\top \mathbf{x}\right) = \text{sgn}(\mathbf{v}^\top \mathbf{x})\left(v_1 x_1 + v_2 x_2\right) - \text{sgn}(\mathbf{v}^\top \mathbf{x})v_1 x_1 = \text{sgn}(\mathbf{v}^\top \mathbf{x})v_2 x_2.$$

Then (22) can be simplified to

$$\rho = \mathbb{E}_{\mathbf{x}\sim\mathcal{D}_{\mathbf{w},\mathbf{v}}}\frac{|\mathbf{v}^\top \mathbf{x}| - \text{sgn}(\mathbf{v}^\top \mathbf{x})\left(\overline{\mathbf{w}}^\top \mathbf{v}\right)\left(\overline{\mathbf{w}}^\top \mathbf{x}\right)}{1 + e^{\text{sgn}(\mathbf{v}^\top \mathbf{x})\mathbf{w}^\top \mathbf{x}}} = \mathbb{E}_{\mathbf{x}\sim\mathcal{D}_{\mathbf{w},\mathbf{v}}}\frac{\text{sgn}(\mathbf{v}^\top \mathbf{x})\left(v_2 x_2\right)}{1 + e^{\text{sgn}(\mathbf{v}^\top \mathbf{x})\|\mathbf{w}\|x_1}} = \mathbb{E}_{\mathbf{x}\sim\mathcal{D}_{\mathbf{w},\mathbf{v}}}g(\mathbf{x})$$

where $g(\mathbf{x}) = \frac{\text{sgn}(\mathbf{v}^\top \mathbf{x})(v_2 x_2)}{1 + e^{\text{sgn}(\mathbf{v}^\top \mathbf{x})\|\mathbf{w}\|x_1}}$. Next, we analyze this expection.

If $|v_1 x_1| > |v_2 x_2|$, then $\text{sgn}(\mathbf{v}^\top \mathbf{x}) = \text{sgn}(v_1 x_1)$, and

$$g(x_1, x_2) + g(x_1, -x_2) = 0.$$

If $|v_2 x_2| \geq |v_1 x_1|$, then $\text{sgn}(\mathbf{v}^\top \mathbf{x}) = \text{sgn}(v_2 x_2)$, and

$$g(x_1, x_2) + g(x_1, -x_2) = \frac{|v_2 x_2|}{1 + e^{\|\mathbf{w}\|x_1}} + \frac{|v_2 x_2|}{1 + e^{-\|\mathbf{w}\|x_1}} = |v_2 x_2| \geq 0.$$

Therefore, we have $g(x_1, x_2) + g(x_1, -x_2) = |v_2 x_2|\mathbb{1}_{\{|v_2 x_2| \geq |v_1 x_1|\}}$. Further due to the spherical symmetry of the distribution $\mathcal{D}$, the marginal distribution $\mathcal{D}_{\mathbf{w},\mathbf{v}}$ satisfies the following property: the probability density at any two points $(x_1, x_2)$ and $(x_1, -x_2)$ are the same. Then

$$\begin{aligned}
\rho &= \mathbb{E}_{\mathbf{x}\sim\mathcal{D}_{\mathbf{w},\mathbf{v}}}g(x_1, x_2) = \frac{1}{2}\mathbb{E}_{\mathbf{x}\sim\mathcal{D}_{\mathbf{w},\mathbf{v}}}|v_2 x_2|\mathbb{1}_{\{|v_2 x_2| \geq |v_1 x_1|\}} \\
&= \frac{|v_2|}{4\pi}\int_0^\infty r \int_{|\tan\theta(\mathbf{w},\mathbf{v})\cdot\tan\theta|\geq 1} |\sin\theta|d\theta dF(r) \\
&= \frac{\sin^2\theta(\mathbf{w},\mathbf{v})}{\pi}\int_0^\infty r dF(r) \\
&= \frac{c_0 \sin^2\theta(\mathbf{w},\mathbf{v})}{\pi},
\end{aligned}$$

where $F(r)$ is the cumulative distribution function of $r := \|\mathbf{x}\|$ and $c_0 = \mathbb{E}_{\mathbf{x}\sim\mathcal{D}_{\mathbf{w},\mathbf{v}}}\|\mathbf{x}\| = \int_0^\infty r dF(r)$.

Next, we prove Eq. (6). If $\theta(\mathbf{w}, \mathbf{v}) = 0$, then Eq. (6) holds from (ii) in Proposition 3. From now on, we assume $\theta(\mathbf{w}, \mathbf{v}) \neq 0$. Denote

$$\hat{\mathbf{u}} := \frac{c_0}{\pi}\mathbf{v} + \nabla L(\mathbf{w}), \quad \hat{\mathbf{W}} := I_d - \overline{\mathbf{w}}\,\overline{\mathbf{w}}^\top. \tag{23}$$

Based on Eq. (5), i.e, the first result in Lemma 2, we obtain

$$-\mathbf{v}^\top\left(I_d - \overline{\mathbf{w}}\,\overline{\mathbf{w}}^\top\right)\nabla L(\mathbf{w}) = \frac{c_0}{\pi}\mathbf{v}^\top\left(I_d - \overline{\mathbf{w}}\,\overline{\mathbf{w}}^\top\right)\mathbf{v}.$$

$$\implies 0 = \mathbf{v}^\top\left(I_d - \overline{\mathbf{w}}\,\overline{\mathbf{w}}^\top\right)\left(\frac{c_0}{\pi}\mathbf{v} + \nabla L(\mathbf{w})\right) \stackrel{(23)}{=} \mathbf{v}^\top\hat{\mathbf{W}}\hat{\mathbf{u}}.$$

Therefore

$$\hat{\mathbf{W}}\hat{\mathbf{u}} \perp \mathbf{v}. \tag{24}$$

We also have

$$\mathbf{w}^\top\hat{\mathbf{W}} = 0 \implies \mathbf{w}^\top\hat{\mathbf{W}}\hat{\mathbf{u}} = 0 \implies \hat{\mathbf{W}}\hat{\mathbf{u}} \perp \mathbf{w}. \tag{25}$$

Moreover, we notice that $\nabla L(\mathbf{w}) \in \text{span}\{\mathbf{w}, \mathbf{v}\}$, thus $\hat{\mathbf{u}} \in \text{span}\{\mathbf{w}, \mathbf{v}\}$, which further implies

$$\hat{\mathbf{W}}\hat{\mathbf{u}} \in \text{span}\{\mathbf{w}, \mathbf{v}\}. \tag{26}$$

Combining Eqs. (24), (25) and (26), we obtain $\hat{\mathbf{W}}\hat{\mathbf{u}} = 0$, i.e.,

$$\left(I_d - \overline{\mathbf{w}}\,\overline{\mathbf{w}}^\top\right)\left(\frac{c_0}{\pi}\mathbf{v} + \nabla L(\mathbf{w})\right) = \mathbf{0}.$$

Thus we have proved Eq. (6). $\qquad\square$

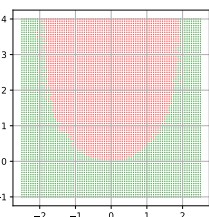

Figure 2: Verification of $\text{sgn}(g(\mathbf{w}))$ when $\mathbf{x} \sim \mathcal{U}(\mathcal{S}^1)$ and $\mathbf{v} = (0, 1)^\top$. The sign of $g(\mathbf{w})$ at each $\mathbf{w}$ is estimated by 1000 random samples. "Red" means positive sign while "Green" is negative sign.

## A.5  Proof of Lemma 3

Proof: Proof of (i): The first proposition follows from

$$g(\mathbf{w}) = -\mathbf{w}^\top \nabla L(\mathbf{w}) = \mathbb{E}_{\mathbf{x} \sim \mathcal{D}} \frac{y(\mathbf{x})\mathbf{w}^\top \mathbf{x}}{1 + e^{y(\mathbf{x})\mathbf{w}^\top \mathbf{x}}}.$$

Use the fact $x/(1 + e^x) < 0.3$, then $g(\mathbf{w}) \le 0.3$.

Proof of (ii) and (iii).

$$g(\mathbf{w}) \overset{(20)}{=} \mathbb{E}_{\mathbf{x} \sim \mathcal{D}_{\mathbf{w},\mathbf{v}}} \frac{y(\mathbf{x})\mathbf{w}^\top \mathbf{x}}{1 + e^{y(\mathbf{x})\mathbf{w}^\top \mathbf{x}}}. \tag{27}$$

Using the spherically symmetric assumption, $\mathcal{D}_{\mathbf{w},\mathbf{v}}$ has the same distribution as $\mathcal{D}_2$, which is also spherically symmetric. Without loss of generality, we can assume $\mathbf{v} = (1, 0), \overline{\mathbf{w}} = (\bar{w}_1, \bar{w}_2), w_2 > 0$ (i.e., $w_i = v_i = 0, i \ge 3$). Then

$$g(\mathbf{w}) \overset{(27)}{=} \mathbb{E}_{\mathbf{x} \sim \mathcal{D}_2} \frac{\text{sgn}(x_1)\|\mathbf{w}\|\overline{\mathbf{w}}^\top \mathbf{x}}{1 + e^{\text{sgn}(x_1)\|\mathbf{w}\|\overline{\mathbf{w}}^\top \mathbf{x}}}. \tag{28}$$

Note that $\overline{\mathbf{w}}^\top \mathbf{x} = \bar{w}_1 x_1 + \sqrt{1 - \bar{w}_1^2} x_2$, which only related to $\bar{w}_1 = \cos\theta(\mathbf{w}, \mathbf{v})$. Hence, $g(\mathbf{w})$ is a function only decided by $\|\mathbf{w}\|$ and $\theta(\mathbf{w}, \mathbf{v})$ from Eq. (28).

Now we reuse the spherically symmetric assumption, and assume $\overline{\mathbf{w}} = (1, 0), \mathbf{v} = (v_1, v_2)$. Then $|v_2/v_1| = |\tan\theta(\mathbf{w}, \mathbf{v})|$, and

$$\frac{g(\mathbf{w})}{\|\mathbf{w}\|} \overset{(27)}{=} \mathbb{E}_{\mathbf{x} \sim \mathcal{D}_2} h(x_1, x_2), \text{ where } h(x_1, x_2) = \frac{\text{sgn}(\mathbf{v}^\top \mathbf{x})x_1}{1 + e^{\text{sgn}(\mathbf{v}^\top \mathbf{x})\|\mathbf{w}\|x_1}}. \tag{29}$$

Case 1. If $|v_1 x_1| > |v_2 x_2|$, i.e., $|x_1| > |x_2 \cdot \tan\theta(\mathbf{w}, \mathbf{v})|$, then $\text{sgn}(\mathbf{v}^\top \mathbf{x}) = \text{sgn}(v_1 x_1)$,

$$h(x_1, x_2) + h(x_1, -x_2) = \frac{2\,\text{sgn}(v_1)|x_1|}{1 + e^{\|\mathbf{w}\|\text{sgn}(v_1)|x_1|}} > 0, \text{ if } x_1 \neq 0.$$

Case 2. If $|v_2 x_2| \ge |v_1 x_1|$, i.e., $|x_1| \le |x_2 \cdot \tan\theta(\mathbf{w}, \mathbf{v})|$, then $\text{sgn}(\mathbf{v}^\top \mathbf{x}) = \text{sgn}(v_2 x_2)$,

$$h(x_1, x_2) + h(x_1, -x_2) = \frac{1 - e^{\|\mathbf{w}\|x_1}}{1 + e^{\|\mathbf{w}\|x_1}} x_1 < 0, \text{ if } x_1 \neq 0.$$

The marginal distribution $\mathcal{D}_{\mathbf{w},\mathbf{v}}$ satisfies the following property: the probability density at any two points $(x_1, x_2)$ and $(x_1, -x_2)$ are the same. Combining Case 1 and Case 2, we could obtain

$$\frac{g(\mathbf{w})}{\|\mathbf{w}\|} = \underbrace{\mathbb{E}_{\mathbf{x} \sim \mathcal{D}_2} \frac{\text{sgn}(v_1)|x_1|}{1 + e^{\|\mathbf{w}\|\text{sgn}(v_1)|x_1|}} \mathbb{1}_{\{|x_1| > |x_2 \cdot \tan\theta|\}}}_{J_1} + \underbrace{\mathbb{E}_{\mathbf{x} \sim \mathcal{D}_2} \frac{1 - e^{\|\mathbf{w}\|x_1}}{1 + e^{\|\mathbf{w}\|x_1}} \cdot \frac{x_1}{2} \mathbb{1}_{\{|x_1| \le |x_2 \cdot \tan\theta|\}}}_{J_2},$$

$$\tag{30}$$

where $v_1 = \cos\theta$ with $\theta = \theta(\mathbf{w}, \mathbf{v})$.

Proof of (ii): When $\theta(\mathbf{w}, \mathbf{v}) > \pi/2$, then $v_1 < 0$, $J_1 < 0$ and $J_2 < 0$, showing that $g(\mathbf{w}) < 0$.

Proof of (iii): When $\theta(\mathbf{w}, \mathbf{v}) \leq \pi/2$, i.e., $v_1 \geq 0$. Then $J_1 \geq 0$ and $J_2 < 0$. Note that $g(\mathbf{w})$ is a function only decided by $\|\mathbf{w}\|$ and $\theta(\mathbf{w}, \mathbf{v})$ from the above argument. We redefine $g(\mathbf{w}) = g(\|\mathbf{w}\|, \theta)$ with $\theta = \theta(\mathbf{w}, \mathbf{v})$. Denoting $u(\mathbf{x}) := \frac{|x_1|}{1 + e^{\|\mathbf{w}\| \cdot |x_1|}} - \frac{1 - e^{\|\mathbf{w}\| x_1}}{1 + e^{\|\mathbf{w}\| x_1}} \cdot \frac{x_1}{2}$, $u(\mathbf{x}) > 0$ if $x_1 \neq 0$. Then for $\theta \leq \pi/2$, and $\theta > \delta > 0$, and some $R > r > 0$,

$$
\begin{aligned}
\frac{g(\|\mathbf{w}\|, \theta - \delta) - g(\|\mathbf{w}\|, \theta)}{\|\mathbf{w}\|} &= \mathbb{E}_{\mathbf{x} \sim \mathcal{D}_2} u(\mathbf{x}) \cdot \mathbb{1}_{\{|x_2 \cdot \tan \theta| \geq |x_1| > |x_2 \cdot \tan(\theta - \delta)|\}} \\
&= 2 \cdot \mathbb{E}_{\mathbf{x} \sim \mathcal{D}_2} u(\mathbf{x}) \cdot \mathbb{1}_{\{x_2/x_1 = \tan \alpha, \pi/2 - \theta \leq \alpha < \pi/2 - (\theta - \delta)\}} \\
&\geq 2 \cdot \mathbb{E}_{\mathbf{x} \sim \mathcal{D}_2} u(\mathbf{x}) \cdot \mathbb{1}_{\{x_2/x_1 = \tan \alpha, \pi/2 - \theta \leq \alpha < \pi/2 - (\theta - \delta)\}} \cdot \mathbb{1}_{\{r^2 \leq x_1^2 + x_2^2 \leq R^2\}} \\
&\geq 2 \cdot \mathbb{E}_{\mathbf{x} \sim \mathcal{D}_2} \epsilon \cdot \mathbb{1}_{\{x_2/x_1 = \tan \alpha, \pi/2 - \theta \leq \alpha < \pi/2 - (\theta - \delta)\}} = \frac{2\delta\epsilon}{\pi},
\end{aligned}
\tag{31}
$$

where we adopt

$$
\epsilon = \min_{\substack{r^2 \leq x_1^2 + x_2^2 \leq R^2 \\ x_2/x_1 = \tan \alpha \\ \pi/2 - \theta \leq \alpha \leq \pi/2 - (\theta - \delta)}} \frac{|x_1|}{1 + e^{\|\mathbf{w}\| \cdot |x_1|}} - \frac{1 - e^{\|\mathbf{w}\| x_1}}{1 + e^{\|\mathbf{w}\| x_1}} \cdot \frac{x_1}{2} > 0,
$$

since $\pi/2 - (\theta - \delta) < \pi/2$, and $\epsilon$ is related to $R, r, \|\mathbf{w}\|$. Hence, we obtain

$$
\frac{\partial g(\|\mathbf{w}\|, \theta)}{\partial \theta} = \lim_{\delta \to 0} \frac{g(\|\mathbf{w}\|, \theta) - g(\|\mathbf{w}\|, \theta - \delta)}{\delta} \leq -\frac{2\epsilon}{\pi} \|\mathbf{w}\| < 0.
$$

We conclude

$$
\frac{\partial g(\mathbf{w})}{\partial \theta(\mathbf{w}, \mathbf{v})} < 0.
$$

Proof of (iv): We fix $\theta(\mathbf{w}, \mathbf{v})$ or $\overline{\mathbf{w}}$, and consider

$$
\overline{g}(r) := \frac{g(\mathbf{w})}{\|\mathbf{w}\|} = \mathbb{E}_{\mathbf{x} \sim \mathcal{D}_2} \frac{y(\mathbf{x}) \overline{\mathbf{w}}^\top \mathbf{x}}{1 + e^{y(\mathbf{x}) r \overline{\mathbf{w}}^\top \mathbf{x}}}.
$$

Then when $r \to 0$, we obtain

$$
\overline{g}(0) = \lim_{r \to 0} \overline{g}(r) = \frac{1}{2} \mathbb{E}_{\mathbf{x} \sim \mathcal{D}_2} y(\mathbf{x}) \overline{\mathbf{w}}^\top \mathbf{x} = \frac{c_0 \cos \theta(\mathbf{w}, \mathbf{v})}{\pi}.
$$

In addition,

$$
\left| \frac{\partial \overline{g}(r)}{\partial r} \right| = \left| \mathbb{E}_{\mathbf{x} \sim \mathcal{D}_2} - \frac{\left( \overline{\mathbf{w}}^\top \mathbf{x} \right)^2 e^{y(\mathbf{x}) r \overline{\mathbf{w}}^\top \mathbf{x}}}{\left( 1 + e^{y(\mathbf{x}) r \overline{\mathbf{w}}^\top \mathbf{x}} \right)^2} \right| \leq \frac{1}{4} \mathbb{E}_{\mathbf{x} \sim \mathcal{D}_2} \left( \overline{\mathbf{w}}^\top \mathbf{x} \right)^2 = \frac{1}{8} \mathbb{E}_{\mathbf{x} \sim \mathcal{D}_2} \|\mathbf{x}\|^2.
$$

Then $\overline{g}(r)$ is $L$-Lipschitz continuous with $L := \frac{1}{8} \mathbb{E}_{\mathbf{x} \sim \mathcal{D}_2} \|\mathbf{x}\|^2$, hence when $r \leq \frac{c_0 |\cos \theta(\mathbf{w}, \mathbf{v})|}{2\pi L}$, we obtain

$$
|\overline{g}(r) - \overline{g}(0)| \leq rL \leq \frac{c_0 |\cos \theta(\mathbf{w}, \mathbf{v})|}{2\pi} \Rightarrow \overline{g}(r) \cdot \overline{g}(0) > 0 \Rightarrow g(\mathbf{w}) \cos \theta(\mathbf{w}, \mathbf{v}) > 0.
$$

$\square$

## A.6 Proof of Lemma 1

**Lemma 9** *(Restatement of Lemma 1) Denote $g(\mathbf{w}) = \frac{1}{2} \cdot \frac{\partial \|\mathbf{w}(t)\|^2}{\partial t}$, and denote $T = \inf\{t : \frac{\partial \|\mathbf{w}(t)\|^2}{\partial t} \geq 0\}$. Under Assumption 1, we have $T < \infty$, and*

$$
g(\mathbf{w}(t)) = \frac{1}{2} \cdot \frac{\partial \|\mathbf{w}(t)\|^2}{\partial t} \begin{cases} \geq 0, & \forall\, t \geq T \\ < 0, & \forall\, t < T. \end{cases}
\tag{32}
$$

**Proof of Lemma 1**: The proof will be divided into three steps.

**Step 1**: We first present an auxiliary lemma, whose proof is given in Appendix A.6.1.

**Lemma 10** *Consider a differentiable function $h(t), t \geq 0$. Suppose for any $t_0$ such that $h(t_0) = 0$, we must have $h'(t_0) > 0$; in other words,*

$$h(t_0) = 0 \Rightarrow h'(t_0) > 0. \tag{33}$$

*Denote $T = \inf\{t : h(t) \geq 0\}$. Then*

$$h(t) \begin{cases} \geq 0, & \forall \, t \geq T \\ < 0, & \forall \, t < T. \end{cases} \tag{34}$$

**Step 2**: Verify that $h(t) := g(\mathbf{w}(t))$ satisfies the condition of the auxiliary lemma.

More precisely, we will show $h(t) := g(\mathbf{w}(t))$ satisfies Eq. (33), i.e., $h(t_0) = 0$ implies $h'(t_0) > 0$.

Suppose $h(t_0) = 0$. According to Lemma 3 (ii) that $g(\mathbf{w}) < 0$ if $\theta(\mathbf{w}, \mathbf{v}) > \pi/2$, we obtain that $g(\mathbf{w}(t_0)) = 0$ implies

$$\theta(\mathbf{w}(t_0), \mathbf{v}) \leq \pi/2. \tag{35}$$

Recall that Lemma 3 (iii) states that $\theta(\mathbf{w}, \mathbf{v}) \leq \pi/2$ implies $\partial g(\mathbf{w})/\partial \theta(\mathbf{w}, \mathbf{v}) < 0$. By Eq. (35)

$$\left. \frac{\partial g(\mathbf{w})}{\partial \theta} \right|_{\mathbf{w} = \mathbf{w}(t_0)} < 0. \tag{36}$$

From Lemma 2 and Eq. (4), we have

$$\frac{\partial \cos \theta(t)}{\partial t} = \frac{c_0}{\pi \|\mathbf{w}(t)\|} \cdot \sin^2 \theta(t), \forall t \geq 0.$$

As mentioned in Remark 2, we can assume the $\|\mathbf{w}(t)\| \neq 0, \forall t \geq 0$. Then by Lemma 11, we obtain $\partial \cos \theta(t)/\partial t > 0$, thus

$$\frac{\partial \theta(t)}{\partial t} < 0, \forall \, t \geq 0. \tag{37}$$

As discussed in the Proof of Lemma 3, $g(\mathbf{w}(t))$ can be viewed as a function only decided by $\|\mathbf{w}(t)\|$ and $\theta(t)$. Then by the chain rule we have

$$h'(t_0) = \frac{\partial g(\mathbf{w}(t_0))}{\partial t} = \frac{\partial g(\mathbf{w}(t_0))}{\partial \|\mathbf{w}\|} \cdot \frac{\partial \|\mathbf{w}(t_0)\|}{\partial t} + \frac{\partial g(\mathbf{w}(t_0))}{\partial \theta} \cdot \frac{\partial \theta(t_0))}{\partial t}$$

$$= \frac{\partial g(\mathbf{w}(t_0))}{\partial \theta} \cdot \frac{\partial \theta(t_0))}{\partial t} > 0.$$

Here the second equality follows from $\frac{\partial \|\mathbf{w}(t_0)\|^2}{\partial t} = 2g(\mathbf{w}(t_0)) = 0$ implies $\|\mathbf{w}(t_0)\| \cdot \frac{\partial \|\mathbf{w}(t_0)\|}{\partial t} = 0$ and the assumption that $\|\mathbf{w}(t)\| \neq 0, \forall t \geq 0$ by Remark 2; the final inequality is due to Eqs. (36) and (37). Therefore we have proved the condition Eq. (33).

Then by Lemma 10, we immediately obtain Eq. (32).

**Step 3**: prove $T < \infty$. We prove by contradiction.

Assume the contrary, that $T = \infty$, then $\partial \|\mathbf{w}(t)\|^2/\partial t \leq 0, \forall t \geq 0$. This implies $\|\mathbf{w}(t)\| \leq \|\mathbf{w}(0)\|, \forall t \geq 0$. However, $\|\mathbf{w}(t)\|$ is unbounded by (iii) in Proposition 2, a contradiction. Thus the assumption does not hold, and we must have $T < \infty$. This finishes the proof. $\qquad \square$

### A.6.1 Proof of Auxiliary Lemmas

**Proof idea of Lemma 10**: The condition Eq. (33) says that whenever the function value reaches 0, it has to "bounce up" (strictly increasing at that point). As a result, once the function value goes above 0, it will never drop down below 0.

**Proof of Lemma 10**: If $T = \infty$, i.e., $h(t) < 0, \forall t \geq 0$, then Eq. (34) automatically holds. From now on, we assume $T < \infty$. By the continuity of $h$, we have $h(T) = 0$; then by Eq. (33), we have $h'(T) > 0$.

We prove Eq. (34) by contradiction. Assume the contrary, that there exists $t_0 > T$, such that $h(t_0) < 0$. Since $h'(T) > 0$, there exists $\epsilon < t_0 - T$ such that $h(T + \epsilon) > 0$. In the interval

$[T + \epsilon, t_0]$, the left end has positive value and the right end has negative value. Thus there exists at least one root. Denote the set of all roots in this interval as $K$, and denote $t_m := \sup\{t : t \in K\}$. By the continuity of $h$, we must have $t_m \in K$ (otherwise, assume $t_m \notin K$, then there is a sequence of roots converging to $t_m$, implying $t_m$ is also a root and thus $t_m \in K$, a contradiction). Then $h(t_m) = 0$ and, by Eq. (33), $h'(t_m) > 0$. By the same argument as before, there exists a certain $\epsilon_m < t_0 - t_m$ such that $h(t_m + \epsilon_m) > 0$, which further implies there is a root $t_p$ in the interval $[t_m + \epsilon_m, t_0]$. This is a contradiction to the definition that $t_m$ is the maximum root on $[T + \epsilon, t_0]$. This contradiction means that there does not exist any $t_0 > T$ such that $h(t_0) < 0$; in other words, $h(t) \geq 0, \forall t > T$. $\qquad \square$

**Lemma 11** *Suppose $f : [0, \infty) \to (0, \infty)$ be a continuous function. Consider the initial value problem*

$$\theta(0) = \theta_0, \frac{\partial \cos \theta(t)}{\partial t} = f(t) \cdot \sin^2 \theta(t), \forall t \geq 0. \tag{38}$$

*If $\theta_0 \neq 0, \pi$, then $\theta(t) \neq 0, \pi$ and $\frac{\partial \cos \theta(t)}{\partial t} > 0, \forall t \geq 0$.*

Proof: The solution to Eq. (38) is

$$\frac{1}{2} \ln \frac{1 + \cos \theta(t)}{1 - \cos \theta(t)} - \frac{1}{2} \ln \frac{1 + \cos \theta_0}{1 - \cos \theta_0} = \int_0^t f(s) ds < \infty.$$

Since $\theta_0 \neq 0, \pi$, then $-\infty < \ln \frac{1 + \cos \theta_0}{1 - \cos \theta_0} < \infty$. Thus $-\infty < \ln \frac{1 + \cos \theta(t)}{1 - \cos \theta(t)} < \infty$, leading to $\theta(t) \neq 0, \pi$. Hence $\sin \theta(t) \neq 0$, showing that $\frac{\partial \cos \theta(t)}{\partial t} > 0$. $\qquad \square$

### A.7 Proofs of Theorem 1

Proof: When $t \leq T$, by Lemma 2, we have

$$\frac{\partial}{\partial t} \cos(\theta(t)) = \frac{1}{\|\mathbf{w}(t)\|} \cdot \frac{c_0 \sin^2 \theta(t)}{\pi} \geq \frac{1}{\|\mathbf{w}(0)\|} \frac{c_0 \sin^2 \theta(t)}{\pi}.$$

We obtain

$$\frac{1}{2} \ln \frac{1 + \cos \theta(t)}{1 - \cos \theta(t)} - \frac{1}{2} \ln \frac{1 + \cos \theta(t_0)}{1 - \cos \theta(t_0)} \geq \frac{c_0}{\|\mathbf{w}(0)\|\pi} t.$$

Thus

$$\cos \theta(t) \geq 1 - \frac{2}{e^{A_1 t + B_1} + 1}, t \leq T.$$

When $t \geq T$, based on (i) in Lemma 3,

$$\frac{\partial}{\partial t} \|\mathbf{w}(t)\|^2 = 2g(\mathbf{w}(t)) \leq 0.6.$$

This implies $\|\mathbf{w}(t)\|^2 \leq 0.6(t - T) + \|\mathbf{w}(T)\|^2$. Together with Lemma 2, we have

$$\frac{\partial}{\partial t} \cos(\theta(t)) = \frac{1}{\|\mathbf{w}(t)\|} \frac{c_0 \sin^2 \theta(t)}{\pi} \geq \frac{1}{\sqrt{0.6(t - T) + \|\mathbf{w}(T)\|^2}} \frac{c_0 \sin^2 \theta(t)}{\pi}.$$

After taking the integral, we have

$$\frac{1}{2} \ln \frac{1 + \cos \theta(t)}{1 - \cos \theta(t)} - \frac{1}{2} \ln \frac{1 + \cos \theta(T)}{1 - \cos \theta(T)} \geq \frac{2c_0}{0.6\pi} \left( \sqrt{0.6(t - T) + \|\mathbf{w}(T)\|^2} - \|\mathbf{w}(T)\| \right).$$

Thus

$$\cos \theta(t) \geq 1 - \frac{2}{e^{A_2 \sqrt{t - T + C_2} + B_2} + 1}, t \geq T.$$

$\qquad \square$

**Remark 1**: Based on Lemma 1, $\|\mathbf{w}(t)\|$ first decreases when $t \leq T$, then increases when $t \geq T$. Thus the approach for analyzing the phase $t \leq T$ above cannot be applied to the phase $t \geq T$.

**Remark 2**: The convergence rate analysis for the stage $t \geq T$ can be applied to any $t \geq 0$. Anyhow, we choose a different approach for the stage $t < T$ since it leads to a stronger bound of the convergence rate.

## A.8 Proof of Lemma 4

**Lemma 12**
$$\|\mathbf{w}(n+1)\|^2 = \left(\overline{\mathbf{w}}(n)^\top \mathbf{w}(n+1)\right)^2 + \left(\frac{c_0 \eta_n}{\pi}\right)^2 \sin^2 \theta(n).$$

Proof:
$$\begin{aligned}
\|\mathbf{w}(n+1)\|^2 &= \|\mathbf{w}(n)\|^2 - 2\eta_n \mathbf{w}(n)^\top \nabla L(\mathbf{w}(n)) + \eta_n^2 \|\nabla L(\mathbf{w}(n))\|^2 \\
&= \left(\|\mathbf{w}(n)\| - \eta_n \overline{\mathbf{w}}(n)^\top \nabla L(\mathbf{w}(n))\right)^2 + \eta_n^2 \left\|\left(I - \overline{\mathbf{w}}(n)\,\overline{\mathbf{w}}(n)^\top\right) \nabla L(\mathbf{w}(n))\right\|^2 \\
&\overset{(1)}{=} \left(\|\mathbf{w}(n)\| - \eta_n \overline{\mathbf{w}}(n)^\top \nabla L(\mathbf{w}(n))\right)^2 + \left(\frac{c_0 \eta_n}{\pi}\right)^2 \left\|\left(I - \overline{\mathbf{w}}(n)\,\overline{\mathbf{w}}(n)^\top\right) \mathbf{v}\right\|^2 \\
&= \left(\|\mathbf{w}(n)\| - \eta_n \overline{\mathbf{w}}(n)^\top \nabla L(\mathbf{w}(n))\right)^2 + \left(\frac{c_0 \eta_n}{\pi}\right)^2 \sin^2 \theta(n) \\
&= \left(\overline{\mathbf{w}}(n)^\top \mathbf{w}(n+1)\right)^2 + \left(\frac{c_0 \eta_n}{\pi}\right)^2 \sin^2 \theta(n).
\end{aligned}$$

The equality in (1) follows from Lemma 2. $\qquad\square$

Now we turn to the main proof of Lemma 4.

Proof: From (ii) in Lemma 3, when $\cos \theta(n) \le 0$, $-\mathbf{w}(n)^\top \nabla L(\mathbf{w}(n)) \le 0$, leading to
$$\|\mathbf{w}(n+1)\|^2 = \|\mathbf{w}(n)\|^2 - 2\eta_n \mathbf{w}(n)^\top \nabla L(\mathbf{w}(n)) + \eta_n^2 \|\nabla L(\mathbf{w}(n))\|^2 \le \|\mathbf{w}(n)\|^2 + \eta_n^2 c_0^2.$$

From Lemma 12, $\|\mathbf{w}(n+1)\| \ge \left|\overline{\mathbf{w}}(n)^\top \mathbf{w}(n+1)\right| \ge \|\mathbf{w}(n)\| - \eta_n \overline{\mathbf{w}}(n)^\top \nabla L(\mathbf{w}(n))$, then

$$\begin{aligned}
\cos \theta(n+1) - \cos \theta(n) &= \frac{1}{\|\mathbf{w}(n+1)\|} \left(\mathbf{v}^\top \left(\mathbf{w}(n) - \eta_n \nabla L(\mathbf{w}(n))\right) - \|\mathbf{w}(n+1)\| \mathbf{v}^\top \overline{\mathbf{w}}(n)\right) \\
&\ge \frac{1}{\|\mathbf{w}(n+1)\|} \left(\mathbf{v}^\top \left(\mathbf{w}(n) - \eta_n \nabla L(\mathbf{w}(n))\right) - \left(\|\mathbf{w}(n)\| - \eta_n \overline{\mathbf{w}}(n)^\top \nabla L(\mathbf{w}(n))\right) \mathbf{v}^\top \overline{\mathbf{w}}(n)\right) \\
&= \frac{1}{\|\mathbf{w}(n+1)\|} \left(-\eta_n \left(\mathbf{v} - (\overline{\mathbf{w}}(n)^\top \mathbf{v})\overline{\mathbf{w}}(n)\right)^\top \nabla L(\mathbf{w}(n))\right) \\
&= \frac{1}{\|\mathbf{w}(n+1)\|} \frac{c_0 \eta_n \sin^2 \theta(n)}{\pi} \ge \frac{\eta_n}{\pi \sqrt{A + \sum_{i=0}^n \eta_i^2}} \sin^2 \theta(n) > 0.
\end{aligned}$$

Thus $\theta(n)$ decrease when $n$ increase. i.e. $\frac{\pi}{2} \le \theta(n+1) \le \theta(n)$ and $0 \ge \cos \theta(n+1) \ge \cos \theta(n) \ge \cos \theta(0) > -1$. Then we obtain

$$\cos \theta(n+1) - \cos \theta(n) \ge \frac{\eta_n (1 - \cos \theta(n))(1 + \cos \theta(n))}{\pi \sqrt{A + \sum_{i=0}^n \eta_n^2}} \ge (1 - \cos \theta(n)) \frac{B \eta_n}{\sqrt{A + \sum_{i=0}^n \eta_i^2}}.$$

Adjust the terms, we get
$$\left(1 - \frac{B \eta_n}{\sqrt{A + \sum_{i=0}^n \eta_i^2}}\right)(1 - \cos \theta(n)) \ge 1 - \cos \theta(n+1),$$

showing that

$$\ln\left(1 - \cos \theta(n)\right) - \ln\left(1 - \cos \theta(n+1)\right) \ge -\ln\left(1 - \frac{B \eta_n}{\sqrt{A + \sum_{i=0}^n \eta_i^2}}\right) \ge \frac{B \eta_n}{\sqrt{A + \sum_{i=0}^n \eta_i^2}}.$$

Hence,
$$\cos \theta(n) \ge 1 - (1 - \cos \theta(0)) e^{-B S_n^-}.$$

When $S_n^- \to +\infty$, for some finite $T$ steps, $\cos \theta(T) \ge 0$, giving that $\mathbf{v}^\top \mathbf{w}(T) \ge 0$. $\qquad\square$

**Remark 6** *Obviously, $S_n^- < n$, and we list several choices of $\{\eta_n\}_{i=1}^\infty$:*
$$S_n^- = \begin{cases} \Theta(n), & \eta_n = \Theta(q^n), q > 1; \\ \Theta(n^{\min\{\alpha+1, 1/2\}}), & \eta_n = \Theta(n^\alpha), -1 < \alpha, \alpha \ne -1/2; \\ \Theta(\ln(n)), & \eta_n = \Theta(n^{-1}); \\ < \infty, & \eta_n = \Theta(n^\alpha), \alpha < -1. \end{cases}$$

### A.9 Proof of Lemma 5

Proof: Use the upper bound of $\|\mathbf{w}(n+1)\|$ below

$$\|\mathbf{w}(n+1)\|^2 = \|\mathbf{w}(n)\|^2 - 2\eta_n \mathbf{w}(n)^\top \nabla L(\mathbf{w}_n) + \eta_n^2 \|\nabla L(\mathbf{w}_n)\|^2 \leq \|\mathbf{w}(n)\|^2 + 0.6\eta_n + c_0^2 \eta_n^2.$$

Therefore, from Lemma 12, we get

$$
\cos\theta\,(n+1) - \cos\theta\,(n) = \frac{\mathbf{v}^\top \mathbf{w}(n+1) - \|\mathbf{w}(n+1)\| \mathbf{v}^\top \overline{\mathbf{w}}(n)}{\|\mathbf{w}(n+1)\|}
$$

$$
= \frac{\left(\mathbf{v} - \mathbf{v}^\top \overline{\mathbf{w}}(n)\overline{\mathbf{w}}(n)\right)^\top \mathbf{w}(n+1) - \left(\|\mathbf{w}(n+1)\| - \overline{\mathbf{w}}(n)^\top \mathbf{w}(n+1)\right)\mathbf{v}^\top \overline{\mathbf{w}}(n)}{\|\mathbf{w}(n+1)\|}
$$

$$
= \frac{1}{\|\mathbf{w}(n+1)\|}\Bigg(-\eta_n \left(\mathbf{v} - (\overline{\mathbf{w}}(n)^\top \mathbf{v})\overline{\mathbf{w}}(n)\right)^\top \nabla L(\mathbf{w}(n))
$$

$$
- \frac{\|\mathbf{w}(n+1)\|^2 - \left(\overline{\mathbf{w}}(n)^\top \mathbf{w}(n+1)\right)^2}{\|\mathbf{w}(n+1)\| + \overline{\mathbf{w}}(n)^\top \mathbf{w}(n+1)}\cos\theta(n)\Bigg)
$$

$$
= \frac{1}{\|\mathbf{w}(n+1)\|}\left(\frac{c_0\eta_n \sin^2\theta(n)}{\pi} - \frac{c_0^2\eta_n^2 \sin^2\theta(n)\cos\theta(n)/\pi^2}{\|\mathbf{w}(n+1)\| + \overline{\mathbf{w}}_n^\top \mathbf{w}(n+1)}\right)
$$

$$
\overset{(i)}{\geq} \frac{1}{\|\mathbf{w}(n+1)\|}\frac{\delta c_0\eta_n \sin^2\theta(n)}{(1+\delta)\pi} \geq \frac{\delta\eta_n}{(1+\delta)\pi\sqrt{A + \sum_{i=0}^n \eta_i^2 + C\eta_i}}\sin^2\theta(n),
$$

where $(i)$ uses the sufficient convergence condition in Eq. (11). Hence, similarly, we obtain

$$\cos\theta(n) \geq 1 - (1 - \cos\theta(0))\,e^{-BS_n^+}.$$

$\square$

**Remark 7** *Obviously, $S_n^+ < n$, and we list several choices of $\{\eta_n\}_{i=1}^\infty$:*

$$
S_n^+ = \begin{cases}
\Theta(n), & \eta_n = \Theta(q^n), q > 1; \\
\Theta(n^{(\min\{\alpha,0\}+1)/2}), & \eta_n = \Theta(n^\alpha), -1 < \alpha; \\
\Theta(\sqrt{\ln(n)}), & \eta_n = \Theta(n^{-1}); \\
< \infty, & \eta_n = \Theta(n^\alpha), \alpha < -1.
\end{cases}
$$

### A.10 Proof of Theorem 2

Proof: Note that $\|\mathbf{w}(n)\| \geq \|\mathbf{w}(n)\|\cos\theta(n) = \mathbf{v}^\top \mathbf{w}(n)$. From (ii) in Proposition 2 and (iii) in Proposition 3, the right term monotonically increases to infinity. Thus, after finite iterations, the sufficient convergence condition would be satisfied: $\|\mathbf{w}(n_1)\|\cos\theta(n_1) \geq R_1 = \eta_+ c_0 + c_0\eta_+/\pi$. Therefore, we have $\|\mathbf{w}(n)\| \geq \|\mathbf{w}(n)\|\cos\theta(n) \geq \|\mathbf{w}(n_1)\|\cos\theta(n_1) \geq R_1, \forall n > n_1$. Thus, from $\eta_+ \geq \eta_n$, we obtain $\forall n > n_1$,

$$\|\mathbf{w}(n+1)\| + \overline{\mathbf{w}}(n)^\top \mathbf{w}(n+1) \geq 2\left(\|\mathbf{w}(n)\| - \eta_n\|\nabla L(\mathbf{w}(n))\|\right) \geq 2c_0\eta_n \cos\theta(n)/\pi,$$

which satisfies the sufficient convergence condition with $\delta = 1$. Hence, from Lemma 5, we obtain superpolynomial directional convergence of $1 - \cos\theta(n)$ from $n_1$. $\square$

## B Missing Proofs in Section 4

### B.1 Proof of Lemma 7

Proof: From the induced flow on $\mathbf{w}_e$, we obtain

$$
-\left(\mathbf{v} - \left(\overline{\mathbf{w}}_e^\top \mathbf{v}\right)\overline{\mathbf{w}}_e\right)^\top \frac{\partial \mathbf{w}_e(t)}{\partial t}
$$

$$
\overset{(14)}{=} -\left(\mathbf{v} - \left(\overline{\mathbf{w}}_e^\top \mathbf{v}\right)\overline{\mathbf{w}}_e\right)^\top \cdot \|\mathbf{w}_e\|^{2-\frac{2}{g}}\left(\frac{dL^1(\mathbf{w}_e)}{d\mathbf{w}} + (N-1)\overline{\mathbf{w}}_e\overline{\mathbf{w}}_e^\top \frac{dL^1(\mathbf{w}_e)}{d\mathbf{w}}\right)
$$

$$
= -\|\mathbf{w}_e\|^{2-\frac{2}{N}}\left(\mathbf{v} - \left(\overline{\mathbf{w}}_e^\top \mathbf{v}\right)\overline{\mathbf{w}}_e\right)^\top \frac{dL^1(\mathbf{w}_e)}{d\mathbf{w}} = \|\mathbf{w}_e\|^{2-\frac{2}{N}} \cdot \frac{c_0 \sin^2\theta(\mathbf{w}_e, \mathbf{v})}{\pi}.
$$

Therefore we have

$$\frac{\partial \cos\theta(\mathbf{w}_e(t), \mathbf{v})}{\partial t} = \frac{\left(\mathbf{v} - \left(\overline{\mathbf{w}}_e(t)^\top \mathbf{v}\right)\overline{\mathbf{w}}_e(t)\right)^\top}{\|\mathbf{w}_e(t)\|} \cdot \frac{\partial \mathbf{w}_e(t)}{\partial t} = \frac{c_0 \sin^2\theta(\mathbf{w}_e(t), \mathbf{v})}{\pi} \cdot \|\mathbf{w}_e(t)\|^{1 - \frac{2}{N}}.$$

$\square$

## B.2  Proof of Lemma 8

Proof: First, we establish the dynamics of the $(2/N)$-th power of the induced norm. More specifically, we will show that if $\mathbf{w}_e(t) \neq \mathbf{0}$, then

$$\frac{\partial \|\mathbf{w}_e(t)\|^{\frac{2}{N}}}{\partial t} = \mathbb{E}_{\mathbf{x} \sim \mathcal{D}_2} \frac{2y(\mathbf{x})\mathbf{w}_e(t)^\top \mathbf{x}}{1 + e^{y(\mathbf{x})\mathbf{w}_e(t)^\top \mathbf{x}}}. \tag{39}$$

This relation is proved as follows.

$$\begin{aligned}
\frac{\partial \|\mathbf{w}_e(t)\|^{\frac{2}{N}}}{\partial t} &= \frac{2}{N}\|\mathbf{w}_e(t)\|^{\frac{2}{N}-1}\overline{\mathbf{w}}_e(t)^\top \frac{\partial \mathbf{w}_e(t)}{\partial t} \\
&\overset{(14)}{=} -\frac{2}{N}\mathbf{w}_e(t)^\top \left(\frac{dL^1(\mathbf{w}_e(t))}{d\mathbf{w}} + (N-1)\overline{\mathbf{w}}_e(t)\overline{\mathbf{w}}_e(t)^\top \frac{dL^1(\mathbf{w}_e(t))}{d\mathbf{w}}\right) \\
&= -2\mathbf{w}_e(t)^\top \frac{dL^1(\mathbf{w}_e(t))}{d\mathbf{w}} = \mathbb{E}_{\mathbf{x} \sim \mathcal{D}_2} \frac{2y(\mathbf{x})\mathbf{w}_e(t)^\top \mathbf{x}}{1 + e^{y(\mathbf{x})\mathbf{w}_e(t)^\top \mathbf{x}}}.
\end{aligned}$$

We now turn to prove each part of Lemma 8.

Proof of (i): From Eq. (39), we have

$$\frac{\partial \|\mathbf{w}_e(t)\|^{\frac{2}{N}}}{\partial t} = \mathbb{E}_{\mathbf{x} \sim \mathcal{D}_2} \frac{2y(\mathbf{x})\mathbf{w}_e(t)^\top \mathbf{x}}{1 + e^{y(\mathbf{x})\mathbf{w}_e(t)^\top \mathbf{x}}} \geq -2\mathbb{E}_{\mathbf{x} \sim \mathcal{D}_2}\|\mathbf{x}\| \cdot \|\mathbf{w}_e(t)\| = -2c_0\|\mathbf{w}_e(t)\|.$$

This implies

$$\frac{2}{2-N} \cdot \frac{\partial \|\mathbf{w}_e(t)\|^{\frac{2}{N}-1}}{\partial t} \geq -2c_0.$$

Integrate both sides for the range $[0, t]$, we obtain the lower bound:

$$\|\mathbf{w}_e(t)\|^{1-\frac{2}{N}} \geq \left(\|\mathbf{w}_e(0)\|^{\frac{2}{N}-1} + (N-2)c_0 t\right)^{-1}.$$

The upper bound is due to (i) in Lemma 3. More specifically,

$$\frac{\partial \|\mathbf{w}_e(t)\|^{\frac{2}{N}}}{\partial t} = \mathbb{E}_{\mathbf{x} \sim \mathcal{D}_2} \frac{2y\mathbf{w}_e(t)^\top \mathbf{x}}{1 + e^{y\mathbf{w}_e(t)^\top \mathbf{x}}} \leq 0.6 \implies \|\mathbf{w}_e(t)\| \leq \left(\|\mathbf{w}_e(0)\|^{\frac{2}{N}} + 0.6t\right)^{\frac{N}{2}}.$$

Proof of (ii): From the lower bound in Eq. (17), we could see

$$\|\mathbf{w}_e(t)\| > 0, \forall t \geq 0. \tag{40}$$

Note that

$$\begin{aligned}
\frac{\partial \|\mathbf{w}_e(t)\|^2}{\partial t} &= \mathbf{w}_e(t)^\top \frac{\partial \mathbf{w}_e(t)}{\partial t} \overset{(14)}{=} -N\|\mathbf{w}_e(t)\|^{2-\frac{2}{N}} \cdot \mathbf{w}_e(t)^\top \frac{dL^1(\mathbf{w}_e(t))}{d\mathbf{w}} \\
&\overset{(7)}{=} N\|\mathbf{w}_e(t)\|^{2-\frac{2}{N}} \cdot g(\mathbf{w}_e(t)).
\end{aligned} \tag{41}$$

Here $g(\mathbf{w}) = -\mathbf{w}^\top \frac{dL^1(\mathbf{w})}{d\mathbf{w}}$ is the same as the one defined in the analysis of logistic regression. This definition applies to any $\mathbf{w}$, including the points on the trajectory of linear model training as well as deep linear net training. Notice that Lemma 3 is established for $g(\mathbf{w})$ with any $\mathbf{w}$, thus can be used for analyzing both linear model training and deep linear net training.

The rest is quite similar to Step 2 in the proof of Lemma 1 in Appendix A.6. There is a slight difference that we need to show the extra term $N\|\mathbf{w}_e(t)\|^{2-\frac{2}{N}}$ in Eq. (41) compared to $\frac{1}{2} \cdot \frac{\partial \|\mathbf{w}_e(t)\|^2}{\partial t} = g(\mathbf{w}_e(t))$ does not affect the argument.

From the assumption $\frac{\partial \|\mathbf{w}_e(t_0)\|^2}{\partial t} = 0$ and Eq. (41), we obtain $g(\mathbf{w}_e(t_0)) = 0$. Further, by (ii) in Lemma 3 we obtain $\theta(t_0) \leq \pi/2$. Note that $\partial \cos \theta(t)/\partial t > 0$ by Eq. (40), Lemmas 7 and 11, leading to $\partial \theta(t)/\partial t < 0, \forall t \geq 0$.

Moreover, we have $\mathbf{w}_e(t) \in \text{span}\{\mathbf{w}_e(0), \mathbf{v}\}$ since $\nabla L(\mathbf{w}_e(t)) \in \text{span}(\mathbf{w}_e(t), \mathbf{v})$ by Eqs. (20) and (14). Hence we can view $g(\mathbf{w}_e(t))$ as a function only decided by $\|\mathbf{w}_e(t)\|$ and $\theta(t)$. Using (iii) in Lemma 3, we have

$$\frac{\partial g(\mathbf{w}_e(t_0))}{\partial t} = \frac{\partial g(\mathbf{w}_e(t_0))}{\partial \|\mathbf{w}_e\|} \cdot \frac{\partial \|\mathbf{w}_e(t_0)\|}{\partial t} + \frac{\partial g(\mathbf{w}_e(t_0))}{\partial \theta} \cdot \frac{\partial \theta(t_0)}{\partial t} = \frac{\partial g(\mathbf{w}_e(t_0))}{\partial \theta} \cdot \frac{\partial \theta(t_0)}{\partial t} > 0,$$

where we use $\partial \|\mathbf{w}_e(t_0)\| /\partial \theta(t) = 0$ by $g(\mathbf{w}_e(t_0)) = 0$ and Eq. (40). This finishes the proof of the desired property: $\frac{\partial \|\mathbf{w}_e(t_0)\|^2}{\partial t} = 0$ implies $\frac{\partial g(\mathbf{w}_e(t_0))}{\partial t} > 0$. $\qquad \square$

### B.3 Proof of Proposition 4

Form Eq. (16) and $N = 2$, we get $\frac{\partial \cos \theta(t)}{\partial t} = \frac{c_0 \sin^2 \theta(t)}{\pi}$. Integrating from 0 to $t$, we obtain

$$\ln \frac{1 + \cos \theta(t)}{1 - \cos \theta(t)} - \ln \frac{1 + \cos \theta(0)}{1 - \cos \theta(0)} = \frac{2c_0}{\pi} t.$$

Hence

$$\cos \theta(t) = 1 - \frac{2}{C_1 e^{2c_0 t/\pi} + 1}, \quad \text{with } C_1 = \frac{1 + \cos \theta(0)}{1 - \cos \theta(0)}.$$

### B.4 Proof of Lemma 6

Lemma 6 consists of two properties: Eq. (15) and $T < \infty$.

First, Eq. (15) is an immediate consequence of Lemma 10 (given in Appendix A.6) and Part (ii) of Lemma 8. More specifically, Part (ii) of Lemma 8 essentially verifies the condition Eq. (33) in Lemma 10 for $h(t) = g(\mathbf{w}_e(t))$. Thus the conclusion of Lemma 10 holds, i.e, Eq. (15) holds.

Next, we prove $T < \infty$ by contradiction. Assume the contrary, that $T = \infty$, i.e.,

$$\partial \|\mathbf{w}_e(t)\|^2/\partial t < 0, \quad \forall t \geq 0. \tag{42}$$

Then $\|\mathbf{w}_e(t)\|$ converges to a finite non-negative value. There are two possibilities.

**Case 1**: $\|\mathbf{w}_e(t)\| \to 0$. We will derive a contradiction. The intuition is that the origin is a saddle point, thus the gradient flow will not converge to it. A formal argument is the following.

We first claim that $\cos \theta(t) \to 1$ [3]. Using Lemma 7 and the lower bound of $\|\mathbf{w}_e(t)\|$ in Lemma 8, we obtain

$$\frac{1}{2} \ln \frac{1 + \cos \theta(t)}{1 - \cos \theta(t)} - \frac{1}{2} \ln \frac{1 + \cos \theta(0)}{1 - \cos \theta(0)} \geq \frac{1}{(N-2)\pi} \ln \frac{A_1 t + B_1}{B_1}.$$

This implies

$$\cos \theta(t) \geq 1 - \frac{2}{C_1 (A_1 t/B_1 + 1)^\alpha + 1}, \forall t \geq 0. \tag{43}$$

As a consequence, there exists $t_0 > 0$ such that $\cos \theta(t) \geq \delta > 0, \forall t \geq t_0$. Since $\mathbf{w}_e(t)$ converges to origin, by (iv) in Lemma 3, for some $t_1 > t_0$ with small enough $\|\mathbf{w}_e(t_1)\|$, we have $g(\mathbf{w}_e(t_1)) \geq 0$, i.e., $\partial \|\mathbf{w}_e(t_1)\|^2/\partial t \geq 0$. This is a contradiction to Eq. (42).

**Case 2**: $\|\mathbf{w}_e(t)\| \to \epsilon > 0$. According to (43), we have $\cos \theta(t) \to 1$, then from (iii) of Lemma 3, we obtain

$$\lim_{t \to \infty} \frac{\partial \|\mathbf{w}_e(t)\|^2}{\partial t} \overset{(41)}{=} \lim_{t \to \infty} N \|\mathbf{w}_e(t)\|^{2 - \frac{2}{N}} \cdot g(\mathbf{w}_e(t)) \overset{(30)}{=} N \epsilon^{2 - \frac{2}{N}} \cdot \mathbb{E}_{\mathbf{x} \sim \mathcal{D}_2} \frac{\epsilon |x_1|}{1 + e^{\epsilon |x_1|}} > 0. \tag{44}$$

This is again a contradiction to Eq. (42).

For both cases we have derived a contradiction. Thus Eq. (42) cannot hold, i.e., $T < \infty$. $\square$

---

[3]The proof of this claim appears in the proof of Theorem 3 as well. The proof of Theorem 3 utilizes this lemma, thus we cannot directly utilize the conclusion of Theorem 3 in the proof of this lemma. Thus we restate that argument here.