# OpenReview forum: "Faster Directional Convergence of Linear Neural Networks under Spherically Symmetric Data"
_NeurIPS.cc/2021/Conference — NeurIPS 2021 Poster_

### Official Review · Reviewer_JkRT · 2021-07-12

**Rating:** 5
**Confidence:** 3

**Summary:**

The authors study both shallow and deep linear networks under the assumption of spherically symmetrical input data. They show that gradient descent provides a super-polynomial convergence rate in this setting, which does not rely on any assumptions on the margin of the data or over-parameterization of the network. They provide some experiments demonstrating their results when their assumptions exactly hold, as well as with deep non-linear activation networks where their assumptions no longer hold.

**Limitations And Societal Impact:**

Yes.

**Main Review:**

Overall, while the theoretical analysis is sound in this work, the impact of the work is unclear to me. In particular, since there are some restrictive data assumptions upon which the analysis relies, it doesn't appear that the analysis is practically useful for anything. That being said, I appreciate the fact that there is no assumption on the margin or overparameterization.

The purpose of experiments on non-linear networks is not very clear to me. If anything, it demonstrates that the general intuitions on super-polynomial convergence that were derived under different assumptions do not carry over when these assumptions are removed. Therefore there is a question of what practical utility the theory provides to neural networks as they are used in practice.  Furthermore, it is strange to me that only test error is displayed, rather than training error, since the theory makes claims about the convergence of training loss rather than test loss. It would be good for the authors to clarify these experiments a bit more.

Proposition 1: The loss function L(w) is only convex in the shallow linear network case as discussed in Section 3. In the deep linear network case where w := (W_1, ..., W_n), the loss is not convex with this parameterization (though it is convex in w_e := (W_n...W_1)^T). This should be clarified.

This is a small thing, but it seems as though most of the claims in Section 4 apply to N > 2 case, while Section 3 applies to N=1 case. However, the N=2 case is not explicitly discussed. For completeness I think the authors should be clear about this specific case.

There are also grammatical errors scattered throughout the text, which should be corrected. I list some examples below:

Line 31: "However, these work consider" --> "However, these works consider"
Line 233: "The intuitive of a)" --> "The intuition of a)"
Line 235:  "...weight is the multiply of the other..." --> "...weight is the product of the other..."

**Time Spent Reviewing:**

5

---

> ### Author Response · Authors · 2021-08-08
> **Response to Reviewer JkRT**
>
> We thank the reviewer for your responsible review.
>
> **Modified after the reviewer's response**
> Impact: Deep learning theory is still in its infancy. At this stage, people are still exploring different convergence proofs and hope to understand more about the mechanism of the optimization trajectories. This is why there are many papers on studying convergence under different scenarios, such as Shamir'COLT19, [R1], [R2]. They do not have an immediate practical impact, and they are still valuable since they contribute to our understanding of deep learning training.
>       As for our paper, we provide arguably the simplest global convergence proof of deep linear networks (though under spherically symmetric data). Our proof is short and can probably provide much insight into the training process. We believe this work can shed light on future studies of deep learning optimization, which is still largely mysterious.
>
> [R1] Alon Brutzkus and Amir Globerson. Globally optimal gradient descent for a convnet with gaussian inputs. In Proceedings of the 34th International Conference on Machine Learning-Volume 70, pages 605–614. JMLR. org, 2017.
>
> [R2] Alon Brutzkus, Amir Globerson, Eran Malach, and Shai Shalev-Shwartz. SGD learns over- parameterized networks that provably generalize on linearly separable data. In International Con- ference on Learning Representations, 2018
>
> We consider the directional convergence, which is proportional to test accuracy (under separable data and spherically symmetrical input data). Hence, we use test accuracy, instead of training loss. In practice, we care more about accuracy instead of surrogate loss. Thus, we want to directly analyze the directional convergence, instead of training loss like previous works do.
>
> Thank you for your suggestions for paper writing and content.

---

> > ### Comment · Reviewer_JkRT · 2021-08-08
> > **Thanks for the Response**
> >
> > Thank you for the clarification on the directional convergence--I see that I made an error in my previous comment on training error. I have raised my score to a 5, to adjust for this.

---

### Official Review · Reviewer_X2Dr · 2021-07-15

**Rating:** 6
**Confidence:** 2

**Summary:**

The authors study the directional convergence of linear and deep linear networks under a specific setting:  data is separable with zero margin and assumed to be spherically symmetric, and the model is trained by gradient flow/descent with the logit loss. In this setting, the authors improve upon previous results by showing fast directional convergence (superpolynomial). Moreover, unlike previous works the authors are able to analyze the directional dynamics along the entire training process without relying on over parametrization or zero error assumption.

**Limitations And Societal Impact:**

They are adressed

**Main Review:**

Overall the authors present an interesting analysis of directional convergence in a restricted setting, which improves upon previous work. My main concern is with the highly narrow scope for which the results apply, as the authors themselves have pointed out (i'd like to commend the authors for providing an honest description of the limitations of their work).
The technical aspect of the derivation rely heavily on the spherically symmetric distribution assumption (assumption 1 in the paper), absent from previous work , rendering the results somewhat incremental (a more complete theoretical handle of convergence in exchange for a more restrictive setting).

I'd like to add that i am no expert in this particular area, hence i will leave the more in depth technical review for other more qualified reviewers.

question:

The authors relate assumption 1 to the benefit of data augmentation and preprocessing. Can they elaborate on this given that recent papers (Neha S et al*) have pointed to the adverse effects of data whitening as preprocessing? How is data augmentation related?

*Neha S. Wadia, Daniel Duckworth, Samuel S. Schoenholz, Ethan Dyer, Jascha Sohl-Dickstein, "Whitening and second order optimization both make information in the dataset
unusable during training, and can reduce or prevent generalization"

**Time Spent Reviewing:**

4

---

> ### Author Response · Authors · 2021-08-08
> **Response to Reviewer X2Dr**
>
> We thank the reviewer for their review and support. [1] is a really interesting paper, though the setting seems unrelated to ours.
> We state the benefit of data augmentation and preprocessing for producing more data, e.g., multiplying a random orthogonal matrix to each data to make the dataset distributes compactly in the whole space.
>
> [1] Neha S. Wadia, Daniel Duckworth, Samuel S. Schoenholz, Ethan Dyer, Jascha Sohl-Dickstein, "Whitening and second order optimization both make information in the dataset unusable during training, and can reduce or prevent generalization.

---

### Official Review · Reviewer_8tLY · 2021-07-22

**Rating:** 4
**Confidence:** 2

**Summary:**

In this paper the authors present a convergence proof of linear neural networks trained with a logit loss. The analysis is based on previous works of Telgarsky, and is only concerned with binary classification of linearly separable points with 0 margin. In addition the authors make the assumption that the data distribution of the points are isotropic namely, spherically symmetric. With this the authors prove a directional convergence result, that states while the parameters magnitude approaches infinity as seen in previous works with separable data and logit losses,  the direction of the parameters converges to the orthogonal vector to the decision hyperplane. The authors present the result for shallow (1 layer) networks as well as deep linear networks. In addition to convergence the authors observe a trend of decreasing and then increasing parameter magnitudes that they also analyze empirically.



**Limitations And Societal Impact:**

The authors did not fully address the limitation and the implication of their data distribution assumption of spherically symmetric data.

**Main Review:**

I find this paper interesting and provides a nice analysis of a new data distribution setting (0-margin separable data). At the same time, some of the assumptions of this work make it too restrictive and similar to previous art. I find the biggest limitation of this work to be the assumption that the data distribution is isotropic and the development of the theory only with respect to a distribution loss. Namely, a spherically symmetric data assumption is very stringent and is very very far from any interesting data manifold of real data. Even for normalized datasets where the covariance is whitened the assumption that the data is spherically symmetric implies that the dataset contains no directionally specific clusters or variation in density which I find to be very far from any real dataset example. Additionally I recommend the authors spend more time to motivate the key discoveries of this work and their importance, as well as tighten up some of the writing as there are many grammatical issues that occasionally obstruct meaning.

**Time Spent Reviewing:**

2

---

> ### Author Response · Authors · 2021-08-08
> **Response to Reviewer 8tLY**
>
> Thank you for your review.
>
> The main comment is "a spherically symmetric data assumption is very stringent and is very very far from any interesting data manifold of real data. Even for normalized datasets where the covariance has whitened the assumption that the data is spherically symmetric implies that the dataset contains no directionally specific clusters or variation in density which I find to be very far from any real dataset example."
>
> While we agree that there is a gap between the data assumption and real data, we would like to emphasize that making assumptions on data distribution is quite common.
>
> i) The same data assumption is adopted from Shamir'COLT 2019. This work analyzed a single-neuron problem with this assumption.
>
> ii) It includes "Gaussian data" as a special case. Gaussian data are quite common, e.g., [R1].
>
> ii) "Linearly separable" is a common assumption on data, and there are many papers using this assumption; e.g. [R2].
>
> All these papers suffer from "the data assumption is very stringent and is very very far from any interesting data manifold of real data."
>
> We would like to say that our understanding of deep learning training is still very limited. Deep learning is still in its infancy. At this stage, it is necessary to make some compromise in the assumptions. We can apply similar criticism to most of the published papers in theoretical DL.
>
> We believe at the current stage, it may be more proper to judge a theoretical paper by whether it makes an interesting contribution. Of course the assumptions matter, but the strength of assumptions shall be combined with the technical contribution.
>
> Let us reiterate our contribution: we provide arguably the simplest global convergence proof of deep linear networks for spherically symmetric data. Our proof is short and can probably provide much insight to the training process. We believe this work can shed light on future studies of deep learning optimization, which is still largely mysterious.
>
> As for typos and grammar issues: thank you for mentioning this. We will proofread and make sure there are no grammar issues in the next version.
>
> [R1] Ohad Shamir. Exponential convergence time of gradient descent for one-dimensional deep linear neural networks. In COLT, pages 2691–2713. PMLR, 2019.
>
> [R2] Alon Brutzkus, Amir Globerson, Eran Malach, and Shai Shalev-Shwartz. SGD learns over-parameterized networks that provably generalize on linearly separable data. In ICLR, 2018

---

> > ### Comment · Reviewer_8tLY · 2021-09-01
> > **Response to rebuttal**
> >
> > I have read the authors comment arguing that in theoretical DL we are still working with simplified situations and data distribution but still would like to argue that the field has advanced in the past 2.5 years from the cited COLT-2019 etc. and that the assumptions made in this work are especially limiting. The other cited works make either the spherically symmetric assumption or non-negative/positive margin assumption. I find a separable data assumption much more viable then the spherically symmetric assumption, that is quite artificial so afer reading all reviews and responses I decided to keep the score as before.

---

### Official Review · Reviewer_Ce73 · 2021-07-30

**Rating:** 6
**Confidence:** 4

**Summary:**

The paper analyzes the setting of shallow and deep linear neural networks under gradient flow and gradient descent and deduces a superpolynomial directional convergence guarantee.
The result is proven under one assumption: spherically symmetric data distribution. Additionally, in the deep linear networks case, it requires a balancedness condition on the initialization.
Among the existing convergence analysis of binary classification on separable data, the current result diverges by examining the dynamics of minimizing the population logit loss (rather than empirical loss) and tests the case of zero margin separable data. It mitigates some assumptions like prior convergence to zero classification error or overparameterization.
The findings are shown empirically by experiments on both benign and real-world datasets and also partially on non-linear fully-connected ReLU networks.

**Limitations And Societal Impact:**

This work mainly focus on the theoretical aspect of deep neural networks. If there are implication that should be addressed I think they only belong to the impact of deep learning in general and not directly to this work.

**Main Review:**




In my view, the paper addresses a fundamental topic for better understanding the theory behind deep neural networks --- convergence analysis. Even though the setting is rather restrictive, the work takes a step forward in understanding convergence in these kinds of models. Further, it mitigates assumptions of previous works with a similar setup and therefore presents an important contribution in this field.
However, there are several issues with the current version that in my opinion should be treated prior to publication.

First, regarding the problem setup. The spherically symmetric distribution is quite a restrictive assumption. Together with assuming separability, it is worth supporting both with stronger motivation. For instance, giving examples for real-world problems that satisfy or nearly achieve those assumptions or preprocessing methods that can approximate this form of the data.
Furthermore, analyzing the population loss instead of empirical loss is a bit nonstandard. In my opinion, it requires justification since in real-world problems we usually don't have access to the distribution but to a finite set of samples, which is the case that is usually addressed.

Regarding the experiments on MNIST and CIFAR, it is mentioned that: "the datasets do not satisfy our data assumption generally". Despite the fact that the reported results do show a positive effect of the acceleration methods. In my view, the theory presented in the paper cannot provide a valid justification for the phenomenon if the data does not of the correct form of the proofs. Therefore, showing it is a bit out of the topic as I see it. Demonstrating these acceleration methods on datasets that do satisfy the data assumption would be better since the presented theory could support its reasoning.

Overall, the contribution of the paper justifies publication in my opinion. Although I think clarifying the reasoning for the setting and focusing on the direct implications of its results are things that the authors should address prior to publication.

### Post-rebuttal update
After reading the author's response and all other comments, my opinion remained as before. I believe the paper is borderline but tend to accept. In my view, the main drawback is still the limited significance of the results within the setting presented in the paper. Other criteria, however, seem to assess the paper positively, as the technical aspects of the paper and the results are solid. Therefore, I keep my score slightly above the acceptance threshold.

**Time Spent Reviewing:**

7

---

> ### Author Response · Authors · 2021-08-08
> **Response to Reviewer Ce73**
>
> Thank you for giving us constructive feedback.
>
> Note that most of the reviewers focus on the direct implications, we recognize that we do not know the practical effect.
> We just want to understand the behavior of neural networks, though restricted to very simple settings.
>
> We also consider the spherically symmetric distribution is quite a restrictive assumption, which could lead to our angle analysis instead of loss analysis as previous works do.
> There are many works [1-8] that analyze the population loss instead of empirical loss since the loss landscape is much better under benign distribution. Even for empirical loss, many works also assume the data are sampled from good distribution.
> As the reviewer mentioned, it requires justification since in real-world problems we usually don't have access to the distribution but to a finite set of samples, which is the case that is usually addressed.
> However, we need some inductive bias to instruct our theory.
> And such benign data distribution makes the analysis simple and may catch some new understanding.
>
> We run the experiments on MNIST and CIFAR to see the correctness of Eq. (8). Since the datasets do not satisfy our assumption, the results cannot provide a valid justification. However, the initial performance seems correct, this may seem a goal for future work.
>
> [1] Zhou, Mo, Rong Ge, and Chi Jin. "A Local Convergence Theory for Mildly Over-Parameterized Two-Layer Neural Network." COLT. 2021.
>
> [2] Yehudai, Gilad, and Shamir Ohad. "Learning a single neuron with gradient methods." Conference on Learning Theory. PMLR, 2020.
>
> [3] Safran, Itay M., Gilad Yehudai, and Ohad Shamir. "The effects of mild over-parameterization on the optimization landscape of shallow relu neural networks." Conference on Learning Theory. PMLR, 2021.
>
> [4] Zhang, Xiao, et al. "Learning one-hidden-layer relu networks via gradient descent." The 22nd international conference on artificial intelligence and statistics. PMLR, 2019.
>
> [5] Li, Yuanzhi, Tengyu Ma, and Hongyang R. Zhang. "Learning over-parametrized two-layer neural networks beyond ntk." Conference on Learning Theory. PMLR, 2020.
>
> [6] Brutzkus, Alon, and Amir Globerson. "Globally optimal gradient descent for a convnet with gaussian inputs." International conference on machine learning. PMLR, 2017.
>
> [7] Du, Simon, et al. "Gradient descent learns one-hidden-layer cnn: Don’t be afraid of spurious local minima." International Conference on Machine Learning. PMLR, 2018.
>
> [8] Goel, Surbhi, Adam Klivans, and Raghu Meka. "Learning one convolutional layer with overlapping patches." International Conference on Machine Learning. PMLR, 2018.

---

> > ### Comment · Reviewer_Ce73 · 2021-08-29
> > **Thanks for your response**
> >
> > Thanks for the clearifications. The analysis presented in the paper does leverage the understanding of the behavior of neural networks. Yet, it seem that the interest in this setting limits the contribution of this work. After reading all reviews and responses I decided to keep the score as before.

---

### Decision · Program_Chairs · 2021-09-27

**Decision:**

Accept (Poster)

**Comment:**

This paper is borderline.  Reviewers generally agreed that the theoretical analysis (main contribution) is technically solid and delivers strong results, but on the other hand is limited to a very restricted setting of spherically symmetric data distributions.  In addition, several reviewers felt that the empirical section on non-linear neural networks is somewhat disconnected from the theory (which applies to linear neural networks), and the value it adds to the paper is limited.  While I share the aforementioned concerns, I believe the state of deep learning theory is such that strong analyses of restricted settings are still of interest, and therefore recommend publication.  I encourage the authors to reconsider the necessity of some of their experiments and defer to the appendix those that do not significantly contribute to the main message of the paper.